# PEA-DPO: Perception-Enhanced Alignment Direct Preference Optimization for MLLMs Alignment

## Abstract

Direct Preference Optimization (DPO) has emerged as an effective approach for aligning large language models (LLMs) with human preferences. However, its adaptation to multimodal settings remains unexplored. Through representational analysis, we identify a key limitation in multimodal preference optimization, which we term **visual insensitivity**: models often fail to distinguish between images and those with critical visual context removed. Our theoretical analysis further uncovers two manifestations of this problem, namely **across-image insensitivity** and **within-image insensitivity**. To address these challenges, we propose Perception-Enhanced Alignment DPO (PEA-DPO), a framework for multimodal LLMs alignment, which explicitly leverages visual preference signals to overcome visual insensitivity. We empirically demonstrate that PEA-DPO enhances sensitivity to visual context while preserving the language modeling capacity of the base model. We empirically evaluate PEA-DPO across three hallucination benchmarks using multimodal LLMs (MLLMs) of varying scales. Our results demonstrate that PEA-DPO effectively mitigates visual insensitivity, achieves stronger multimodal alignment, and substantially reduces hallucinations.

## 1 Introduction

Aligning multimodal large language models (MLLMs) with human values is crucial for building reliable AI systems that can understand and reason about visual-textual content while generating helpful responses (Yin et al., 2024; Bai et al., 2025; Liu et al., 2024). The prevailing approaches in this field follow the advances established by text-only language model alignment (Pi et al., 2024; Sarkar et al., 2024), applying Direct Preference Optimization (DPO) (Rafailov et al., 2023) and its variants (Fu et al., 2025; Meng et al., 2024; Yang et al., 2025; Lu et al., 2025) to multimodal scenarios. They typically first construct multimodal preference datasets by pairing images with corresponding preferred and dispreferred textual responses, then apply standard DPO loss to align model outputs with human judgments (Li et al., 2023; Xiao et al., 2024; Zhao et al., 2023; Zhou et al., 2024a; Yu et al., 2024a; Deng et al., 2024).

However, directly applying DPO to MLLMs implicitly treats the image as *conditioning* rather than a *target* of preference. As a result, the model can produce responses that are weakly grounded in the visual evidence. We term this failure **visual insensitivity**. Empirically, Fig. 1 shows that the representation distributions of origin images (*i.e.,* chosen images) and their context-reduced counterparts (*i.e.,* rejected images, with critical visual evidence removed) largely overlap for both a base LLaVA model and its DPO-tuned version, indicating poor separation between informative and uninformative visual inputs.

To provide theoretical foundation for these observations, we formalize visual insensitivity by decomposing the model's preference into *textual* and *visual* components. The analysis exposes two failure modes: (i) **across-image insensitivity**, where the model assigns nearly identical preference across distinct images; and (ii) **within-image insensitivity**, where the model fails to discriminate semantically critical cues from irrelevant ones within the same image. Fig. 2 illustrates both phenomena: near-identical responses across different scenes (*e.g.,* outdoor reading vs. picnic) and misrecognition of key objects (*e.g.,* toilet brush vs. generic tool).

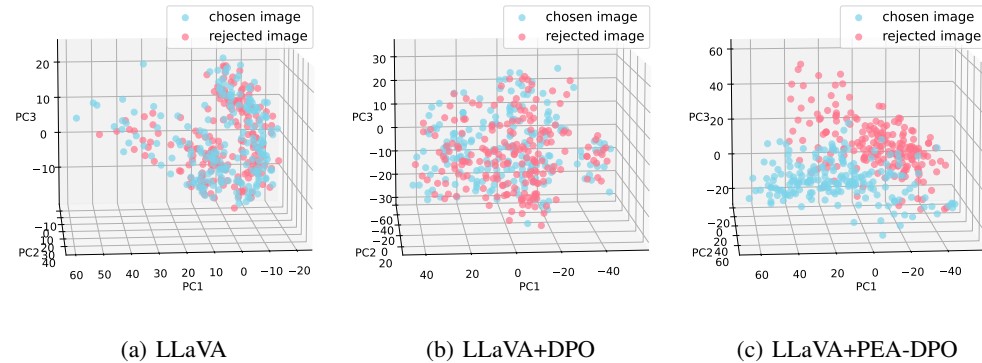

Figure 1: Comparison of representation distributions for different models. Representations are constructed from 200 samples (original images and images with removed critical visual context), using the embedding of the last token from the LLM to represent image semantics.

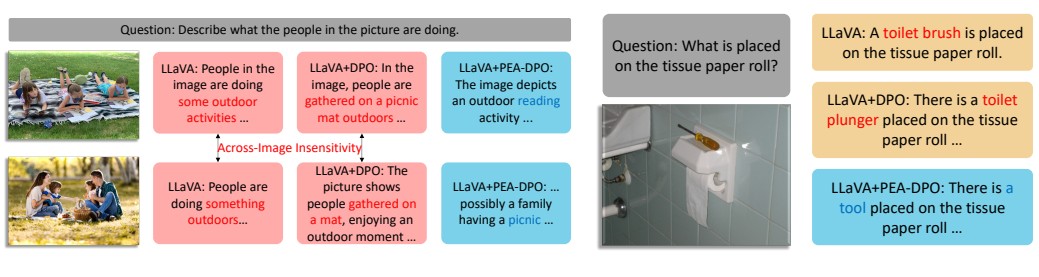

Figure 2: Illustration of two manifestations of visual insensitivity in MLLM alignment. (a) Across-image insensitivity: the model assigns nearly identical preference across distinct images (*e.g.,* outdoor reading vs. picnic), failing to capture discriminative visual context. (b) Within-image insensitivity: the model fails to discriminate semantically critical cues from irrelevant ones within the same image. (*e.g.,* confusing a screwdriver with a toilet brush or a plunger), leading to visually ungrounded responses.

To address these limitations, we propose PEA-DPO, which fundamentally shifts from treating images as static conditioning context to jointly optimizing response quality preferences and visual context preferences. At its core is a dual preference learning framework that optimizes two complementary signals simultaneously: (1) response quality preference, where given the same text-image input, the model learns to distinguish high-quality responses from worse ones (standard multimodal DPO), and (2) visual context preferences, where given the same text but different visual contexts (original vs. context-reduced images), the model should favor responses that properly utilize complete visual evidence. We implement this dual optimization through two key components: (1) Construction of Perception-enhanced Preference Data, wherein we generate candidate images by applying random masks to original images, then leverage CLIP embeddings to identify masked variants that eliminate the most critical visual context, creating meaningful visual grounding preference pairs; and (2) Joint Optimization Objective, which combines both preference learning terms to simultaneously enhance response quality and visual sensitivity, promoting differentiation between different visual contexts (cross-image sensitivity) while directing attention toward semantically critical visual elements within images (within-image sensitivity).

To evaluate the effectiveness of PEA-DPO, we conduct experiments using two sizes of LLaVA-v1.5 models (Liu et al., 2024), with 7B and 13B parameters. Evaluations on MMHal Bench (Sun et al., 2023), Object HalBench (Rohrbach et al., 2018), and AMBER (Wang et al., 2023) demonstrate that PEA-DPO significantly outperforms strong commercial multimodal models such as GPT-4V (Achiam et al., 2023) in multimodal scenarios. Furthermore, for both 7B and 13B models, PEA-DPO achieves SOTA performance on all benchmarks, highlighting its effectiveness and scalability.

## 2 RELATED WORK

Recent research on multimodal large language models (MLLMs) has increasingly focused on aligning textual and visual modalities through preference-based learning. Early approaches extend Direct Preference Optimization (DPO) (Rafailov et al., 2023) from text-only LLMs to multimodal settings, aiming to mitigate hallucinations and improve factual consistency. Notable methods include mDPO (Wang et al., 2024), which introduces conditional preference optimization to handle multimodal inputs, and V-DPO (Xie et al., 2024), which leverages vision-guided signals to enhance the preference model's sensitivity to critical visual content. Similarly, CHiP (Fu et al., 2025) proposes a hierarchical, cross-modal preference optimization framework to better capture multimodal dependencies and reduce spurious correlations between vision and language. Moreover, DAMA (Lu et al., 2025) jointly considers data and model characteristics to improve debiasing in multimodal LLMs. SymPO (Liu et al., 2025b) further enforces theoretical consistency across modalities, ensuring that model outputs remain robust under perturbations in either text or vision inputs. Other lines of work explore preference optimization to enhance robustness and generalization. AdPO (Liu et al., 2025a) applies adversarial preference signals to strengthen multimodal models against input perturbations, while OPA-DPO (Yang et al., 2025) highlights the importance of on-policy data in preventing hallucinations, showing that dynamically collected preference data can outperform static datasets in aligning model behavior. Despite these advancements, a critical issue remains underexplored: the **visual insensitivity** problem in multimodal DPO. This issue manifests in two forms: **Across-Image Insensitivity** and **Within-Image Insensitivity**. We propose a solution to this issue by explicitly incorporating visual preference signals.

## 3 BACKGROUND

### 3.1 BACKGROUND: DIRECT PREFERENCE OPTIMIZATION IN MULTIMODAL SCENARIO

To further improve the performance of MLLMs, RLHF/RLAIF requires a reward model $r(x, y, m)$ that evaluates human preference over a response $y$ given a prompt $x$ and image $m$. The standard learning objective is:

$$\max_{\pi_\theta} \mathbb{E}_{\mathcal{D}_y}[r(x, y, m)] - \beta \mathbb{D}_{\text{KL}}[\pi_\theta(\cdot \mid x, m) \mid\mid \pi_{\text{ref}}(\cdot \mid x, m)], \quad (1)$$

where $\mathcal{D}$ denotes the dataset, with prompts $x$ and images $m$ sampled from the reference policy $\pi_{\text{ref}}$. The term $\mathbb{D}_{\text{KL}}$ is the KL divergence, and $\beta$ controls the strength of regularization. DPO derives a closed-form solution to Eq. 1, revealing that the reward function can be expressed as:

$$r(x, y, m) = \beta \log \frac{\pi_\theta(y \mid x, m)}{\pi_{\text{ref}}(y \mid x, m)} + \beta \log Z(x, m), \quad (2)$$

where $Z(x, m)$ is a partition function depending only on the prompt $x$ and image $m$. Incorporating this into the Bradley–Terry model (Bradley & Terry, 1952), and given a dataset of preference pairs $(y_w \succ y_l)$ under the same $(x, m)$, the optimization objective becomes:

$$\mathcal{L}_{\text{DPO}} = -\mathbb{E}_{\mathcal{D}}[\log \sigma(r(x, y_w, m) - r(x, y_l, m))]$$
$$= -\mathbb{E}_{\mathcal{D}}\left[ \log \sigma\left( \beta \log \frac{\pi_\theta(y_w \mid x, m)}{\pi_{\text{ref}}(y_w \mid x, m)} - \beta \log \frac{\pi_\theta(y_l \mid x, m)}{\pi_{\text{ref}}(y_l \mid x, m)} \right) \right], \quad (3)$$

where $\sigma(\cdot)$ denotes the sigmoid function.

### 3.2 PROBLEM: ACROSS-IMAGE INSENSITIVITY V.S. WITHIN-IMAGE INSENSITIVITY

In this section, we provide a theoretical analysis of visual insensitivity in multimodal DPO, which manifests in two forms: (1) across-image insensitivity and (2) within-image insensitivity.

**Definition 1 (image-based likelihood ratio)** *Inspired by (Gutmann & Hyvärinen, 2010; Oord et al., 2018), we define the image-based likelihood ratio in log form:*

$$\ell_\theta(x, m; y) \triangleq \log \frac{\pi_\theta(y \mid x, m)}{\pi_\theta(y \mid x)}, \quad (4)$$

*which quantifies the relative gain in the plausibility of response $y$ when conditioning on the image $m$ in addition to the prompt $x$.*

**Definition 2 (response-based margin)** *Given a preference data $(x, m, y_w, y_l)$, the response-based margin is defined as follows:*

$$M_\theta^y(x, m) \triangleq \log \pi_\theta(y_w \mid x, m) - \log \pi_\theta(y_l \mid x, m). \tag{5}$$

$M_\theta^y$ quantifies the relative preference of the policy $\pi_\theta$ between the preferred response $y_w$ and the dispreferred response $y_l$. $M_\theta^y$ can be decomposed as:

$$M_\theta^y(x, m) = \underbrace{\left[\log \pi_\theta(y_w \mid x) - \log \pi_\theta(y_l \mid x)\right]}_{\Delta_{\text{text}}^\theta} + \underbrace{\left[\ell_\theta(x, m; y_w) - \ell_\theta(x, m; y_l)\right]}_{\Delta_{\text{vis}}^\theta}, \tag{6}$$

where $\Delta_{\text{vis}}$ measures the extend to which image $m$ reinforces the preference for the preferred response. Given a pair of images $m_w, m_l$, where $m_w$ enables the prompt $x$ to better align with the chosen response $y_w$ than $m_l$.

**Theorem 1 (Across-Image Insensitivity)** *Suppose there exist samples $(x, m_w, m_l, y_w, y_l)$ such that*

$$M_\theta^y(x, m_w) - M_\theta^y(x, m_l) \leq \delta \quad (\delta \to 0), \tag{7}$$

*it implies the relation:*

$$\Delta_{\text{vis}}^\theta(x, m_w) - \Delta_{\text{vis}}^\theta(x, m_l) \leq \delta \quad (\delta \to 0). \tag{8}$$

*Intuitive explanation.* The model cannot effectively distinguish the impact of $m_w$ versus $m_l$ on the responses, a phenomenon we term **Across-Image Insensitivity**.

**Theorem 2 (Within-Image Insensitivity)** *Suppose there exist samples $(x, m_l, y)$ such that*

$$\left|\ell_\theta(x, m_l; y)\right| \leq \varepsilon \quad (\varepsilon \to 0), \tag{9}$$

*and that the model exhibits **Across-Image Insensitivity**. Then the following holds for $m_w$:*

$$\left|\ell_\theta(x, m_w; y_w) - \ell_\theta(x, m_w; y_l)\right| \leq \delta + 2\varepsilon \quad (\delta, \varepsilon \to 0). \tag{10}$$

*Intuitive explanation.* This observation suggests that the model is unable to reliably generate the correct response even when conditioned on $m_w$. We refer to this phenomenon as **Within-Image Insensitivity**. Complete proofs are provided in Appendix B.

## 4 METHOD

In summary, both types of issues arise from visual insensitivity. To address these limitations, we introduce PEA-DPO, which consists of two key components: (1) Construction of Perception-Enhanced Preference Data, where rejected images $m_l$ are generated by removing critical visual context using a CLIP-based approach; and (2) Joint Optimization Objective, which learning response quality preferences and visual sensitivity, simultaneously optimizing over response-level and image-level preferences, thereby aligning with human value preferences while enhancing visual sensitivity.

### 4.1 CONSTRUCTION OF PERCEPTION-ENHANCED PREFERENCE DATA

We begin with the chosen image $m_w$ (*i.e.,* the original image) and apply a random mask of fixed proportion to generate a perturbed image $m_p$:

$$m_p = m_w \odot (1 - \mathbf{P}), \tag{11}$$

where $\mathbf{P}$ denotes a random binary mask and $\odot$ represents the Hadamard product. Repeating this process $n$ times yields a candidate set of perturbed images:

$$\mathcal{M}_p = \{m_p^k\}_{k=1}^n, \tag{12}$$

where $m_p^k$ denoting the $k$-th perturbed image. Since the perturbations are random, each perturbed image retains different portions of the original visual information. To quantify how much visual context is lost relative to the chosen image $m_w$, we compute semantic similarity using CLIP (Radford

Figure 3: Overview of **PEA-DPO**. *Left*: Original text-based preference pairs. *Top*: Construction process of perception-enhanced preference pairs, where critical key visual context is removed from images using a CLIP-based approach. *Right*: The resulting perception-enhanced preference pairs. *Bottom*: Joint optimization text-based and perception-enhanced preferences.

et al., 2021) embeddings. Let $\mathbf{v}_w = f_{\text{CLIP}}(m_w)$ and $\mathbf{v}_p^k = f_{\text{CLIP}}(m_p^k)$ denote the $\ell_2$-normalized embeddings from the CLIP image encoder $f_{\text{CLIP}}(\cdot)$. The similarity score is defined as:

$$s_k = \cos(\mathbf{v}_w, \mathbf{v}_p^k) = \frac{\mathbf{v}_w^\top \mathbf{v}_p^k}{\|\mathbf{v}_w\|_2 \|\mathbf{v}_p^k\|_2}, \qquad (13)$$

where $s_k \in [-1, 1]$ measures semantic similarity. Intuitively, given that all perturbations have equal mask size, a lower similarity score indicates that critical visual context has been removed. Finally, we select the perturbed image with the lowest similarity as the rejected image:

$$m_l = \arg \min_{m_p^k \in \mathcal{M}_p} s_k, \qquad (14)$$

where $m_l$ corresponds to the image with its key visual context removed. Optimizing over the perception-enhanced preference data $(x, m_l, m_w, y_w)$ encourages the model to leverage critical visual cues, thereby improving its ability to generate preferred responses grounded in visual evidence.

## 4.2 Joint Optimization of Response- and Image-Based Preferences

After constructing the Perception-Enhanced Preference Data, we obtain a dataset $\mathcal{D}_v = (x, m_w, m_l, y_w)$. Building upon the standard multimodal DPO, we replace the response quality preference data $\mathcal{D} = (x, m_w, y_w, y_l)$ with $\mathcal{D}_v$ in Eq. 3. This yields Visual context Preference Optimization (VPO) Objective, formulated as:

$$\mathcal{L}_{\text{VPO}} = -\mathbb{E}_{\mathcal{D}_v}[\log \sigma(r(x, y_w, m_w) - r(x, y_w, m_l))]$$
$$= -\mathbb{E}_{\mathcal{D}_v}\left[\log \sigma\left(\beta \log \frac{\pi_\theta(y_w \mid x, m_w)}{\pi_{\text{ref}}(y_w \mid x, m_w)} - \beta \log \frac{\pi_\theta(y_w \mid x, m_l)}{\pi_{\text{ref}}(y_w \mid x, m_l)}\right)\right]. \qquad (15)$$

Combining this with the standard multimodal DPO (*i.e.,* Response quality Preference Optimization, RPO) in Eq. 3, we obtain the PEA-DPO objective:

$$\mathcal{L}_{\text{PEA-DPO}} = \mathcal{L}_{\text{RPO}} + \alpha \cdot \mathcal{L}_{\text{VPO}}$$
$$= -\mathbb{E}_{\mathcal{D}}\left[\log \sigma\left(\beta \log \frac{\pi_\theta(y_w \mid x, m_w)}{\pi_{\text{ref}}(y_w \mid x, m_w)} - \beta \log \frac{\pi_\theta(y_l \mid x, m_w)}{\pi_{\text{ref}}(y_l \mid x, m_w)}\right)\right]$$
$$- \mathbb{E}_{\mathcal{D}_v}\left[\log \sigma\left(\beta \log \frac{\pi_\theta(y_w \mid x, m_w)}{\pi_{\text{ref}}(y_w \mid x, m_w)} - \beta \log \frac{\pi_\theta(y_w \mid x, m_l)}{\pi_{\text{ref}}(y_w \mid x, m_l)}\right)\right], \qquad (16)$$

where $\alpha$ is a weighting hyperparameter. This joint objective enables MLLMs to align with human preferences by simultaneously leveraging textual and critical visual modalities.

To further reduce computational overhead and enable controllable preference optimization, inspired by the design of RePO (Wu et al., 2025) and introduce three modifications to $\mathcal{L}_{\text{PEA-DPO}}$: (1) Replace

log-probability ratios in Eq. 16 with length-normalized log-probabilities, thereby eliminating the need for a reference model; (2) Replace the sigmoid function with a ReLU activation, which filters out trivial data points and prevents overfitting; (3) Introduce target margin $\{\gamma_r, \gamma_m\}$ and remove the temperature parameter $\beta$, enabling a controllable optimization process. The modified PEA-DPO loss function is then given as:

$$
\begin{aligned}
\mathcal{L}_{\text{mPEA-DPO}} =& \mathcal{L}_{\text{mRPO}} + \alpha \cdot \mathcal{L}_{\text{mVPO}} \\
=& \mathbb{E}_{\mathcal{D}}\left\{ \text{ReLU}\left[ -\left( \frac{\log \pi_\theta(y_w|x, m_w)}{|y_w|} - \frac{\log \pi_\theta(y_l|x, m_w)}{|y_l|} - \gamma_r \right) \right] \right\} \\
& + \alpha \cdot \mathbb{E}_{\mathcal{D}_m}\left\{ \text{ReLU}\left[ -\left( \frac{\log \pi_\theta(y_w|x, m_w)}{|y_w|} - \frac{\log \pi_\theta(y_w|x, m_l)}{|y_w|} - \gamma_m \right) \right] \right\},
\end{aligned}
\tag{17}
$$

where $|y|$ denotes the number of tokens in response $y$, and $\{\gamma_r, \gamma_m\}$ are the target reward margins, enforcing a minimum separation between preferred and rejected responses in both response-based preferences $\{y_w, y_l\}$ and image-based preferences $\{m_w, m_l\}$.

## 5 EXPERIMENTS

### 5.1 EXPERIMENTAL SETUP

**Models.** We evaluate mPEA-DPO on two MLLMs with different parameter scales: LLaVA-v1.5-7B and LLaVA-v1.5-13B (Liu et al., 2024), both equipped with a CLIP ViT-L-336px vision encoder. The 7B model is built upon Vicuna-7B as its LLM backbone, while the 13B model utilizes Vicuna-13B as its LLM backbone. Both models are first pretrained on 558K image–text pairs datasets and then fine-tuned on 665K instruction-following instances.

**Training Data.** Following (Lu et al., 2025), we utilize the preference data of the LLaVA-1.5 model released by (Yu et al., 2024b), where the language-based preference is annotated by the open-source LLaVA-NeXT-34B model. Specifically, the dataset comprises 22K preference instances, with 13K images used for training. To construct perception-enhanced preference data, we generate rejected images by masking informative regions in these 13K images.

**Baselines.** We report results across three categories of multimodal alignment approaches, while noting that direct comparison is non-trivial due to differences in base models, preference data, and alignment strategies. Specifically: (1) **Hallucination-specific baselines**: including VCD (Leng et al., 2024), OPERA (Huang et al., 2024), HALC (Jiang et al., 2024), and EOS (Yue et al., 2024b), (2) **RLHF/RLAIF-based baselines**: including POVID (Zhou et al., 2024b), LLaVA-RLHF (Sun et al., 2023), HALVA (Sarkar et al., 2024), RLHF-V (Yu et al., 2024a), HA-DPO (Zhao et al., 2023), HSA-DPO (Xiao et al., 2024), RLAIF-V (Yu et al., 2024b), mDPO (Wang et al., 2024), OPA-DPO (Yang et al., 2025), and DAMA (Lu et al., 2025). (3) **Proprietary baseline**: GPT-4V (Achiam et al., 2023), which we use as a robust reference to compare the performance between open-source and proprietary commercial models. Gemini-2.5-Pro (Comanici et al., 2025), which is also included in our comparison, given its status as a strong and up-to-date commercial baseline.

**Benchmarks.** (1) **Object HalBench** (Rohrbach et al., 2018) is a widely adopted benchmark for evaluating object hallucination, focusing on detailed image descriptions of visual content. Following the protocol in (Yu et al., 2024b), we evaluate on 300 instances and report hallucination rates at both the response-level (CHAIR$_S$) and object-level (CHAIR$_I$). (2) **AMBER** (Wang et al., 2023) provides a multi-dimensional evaluation of hallucination in MLLMs. Using its generative task with 1K samples, we report CHAIR scores, object coverage, hallucination rates, and alignment with human cognition. (3) **MMHal-Bench** (Sun et al., 2023) is a question-answering benchmark comprising 96 image–question pairs across 12 object categories and 8 question types. Following the setup of (Yu et al., 2024b), we assess both overall response quality (scored from zero to six) and hallucination rate, as judged by GPT-4.

**Implementation Details.** In our experiments, we construct perception-enhanced preference data by masking $9\%$ of each image, repeated $n = 30$ times. We employ LLaVA-1.5-7B and LLaVA-1.5-

13B as backbone models, and perform full-parameter fine-tuning for 5,000 steps. Specifically, the batch size is set to 32 for the 7B model and 16 for the 13B model. In our loss formulation Eq. 17, we set the hyperparameters as $\gamma_r = 1.5$, $\gamma_m = 4.5$ and $\alpha = 0.2$. All experiments are conducted using 4 NVIDIA A100 80GB GPUs.

Table 1: **Main results** of LLaVA-v1.5-7B and LLaVA-v1.5-13B trained with different preference optimization objectives. We report overall score (Score) and hallucination rate (Hal.) on MMHal-Bench, CHAIR scores at both response and object levels on Object HalBench, along with CHAIR scores (C.), object coverage (cover.), hallucination rate (Hal.), and cognition (Cog.) on AMBER. The best result for each metric in each group is in **bold**. For completeness, we further report additional results obtained with a range of multimodal LLMs, preference datasets, and training objectives, even though these results are not directly comparable. We have carefully followed the publicly available code and released checkpoints to reproduce these results, aiming to provide a fair comparison. [†] indicates results obtained using the official API, [‡] indicates results reproduced using the authors' released code, and [♯] indicates results produced from the authors' provided checkpoints.

| | MMHalBench | | Object HalBench | | AMBER | | | |
|---|---|---|---|---|---|---|---|---|
| | Score ↑ | Hal. ↓ | CHAIR$_s$ ↓ | CHAIR$_i$ ↓ | C. ↓ | Cover. ↑ | Hal. ↓ | Cog. ↓ |
| GPT-4V (Achiam et al., 2023)[†] | 3.49 | 0.28 | 13.6 | 7.3 | 4.6 | 67.1 | 30.7 | 2.6 |
| Gemini-2.5-pro (Comanici et al., 2025)[†] | 3.55 | 0.28 | 12.0 | 8.2 | 9.5 | 78.0 | 75.1 | 5.2 |
| *7B MLLMs* | | | | | | | | |
| LLaVA-v1.5-7B (Liu et al., 2024) | 2.11 | 0.54 | 53.6 | 25.2 | 7.8 | 51.0 | 36.4 | 4.2 |
| + HACL (Jiang et al., 2024)[‡] | 2.13 | 0.50 | - | - | - | - | - | - |
| + OPERA (Huang et al., 2024)[‡] | 2.15 | 0.54 | 45.1 | 22.3 | - | - | - | - |
| + VCD (Leng et al., 2024)[‡] | 2.12 | 0.54 | 48.8 | 24.3 | - | - | - | - |
| + EOS (Yue et al., 2024b)[‡] | 2.03 | 0.59 | 40.3 | 17.8 | 5.1 | 49.1 | 22.7 | 2.0 |
| + POVID (Zhou et al., 2024b)[‡] | 2.08 | 0.56 | 48.1 | 24.4 | - | - | - | - |
| + LLaVA-RLHF (Sun et al., 2023)[♯] | 1.88 | 0.71 | 58.0 | 15.6 | 9.7 | **53.2** | 46.6 | 5.3 |
| + HA-DPO (Zhao et al., 2023)[♯] | 1.97 | 0.60 | 39.9 | 19.9 | 6.7 | 49.8 | 30.9 | 3.3 |
| + HALVA (Sarkar et al., 2024)[♯] | 2.25 | 0.54 | - | - | 6.6 | 53.0 | 32.2 | 3.4 |
| + mDPO (Wang et al., 2024)[‡] | 2.39 | 0.54 | 35.7 | 9.8 | 4.4 | 52.4 | 24.5 | 2.4 |
| + RLAIF-V (Yu et al., 2024b)[♯] | 3.00 | 0.38 | 16.0 | 3.7 | 3.0 | 50.4 | 16.2 | 1.0 |
| + OPA-DPO (Yang et al., 2025)[♯] | 2.83 | 0.45 | 13.0 | 4.3 | 2.2 | 47.9 | 11.6 | 0.9 |
| + DAMA (Lu et al., 2025)[♯] | 2.76 | 0.41 | 10.3 | 5.9 | 3.0 | 48.3 | 14.8 | 1.2 |
| **+ mPEA-DPO** | **3.02** | **0.36** | **4.3** | **3.2** | **1.9** | 46.7 | **10.3** | **0.6** |
| *13B MLLMs* | | | | | | | | |
| LLaVA-v1.5-13B (Liu et al., 2024) | 2.42 | - | 46.3 | 22.6 | 7.8 | 51.0 | 36.4 | 4.2 |
| + LLaVA-RLHF (Sun et al., 2023)[♯] | 2.27 | 0.64 | 44.7 | 11.8 | 7.7 | 52.3 | 38.6 | 4.0 |
| + RLHF-V (Yu et al., 2024a)[♯] | 2.81 | 0.49 | 12.2 | 7.5 | 6.3 | 46.1 | 25.1 | 2.1 |
| + HALVA (Sarkar et al., 2024)[♯] | 2.58 | 0.45 | - | - | 6.4 | **52.6** | 30.4 | 3.2 |
| + OPA-DPO (Yang et al., 2025)[♯] | 3.07 | 0.39 | 16.33 | 5.5 | **2.4** | 48.3 | **12.8** | **0.8** |
| + DAMA (Lu et al., 2025)[♯] | 2.89 | 0.43 | 7.7 | 4.9 | 3.0 | 50.5 | 14.1 | 0.9 |
| **+ mPEA-DPO** | **3.16** | **0.31** | **4.3** | **2.7** | 2.9 | 50.0 | **12.8** | **0.8** |

## 5.2 MAIN RESULTS

The experimental results of applying mPEA-DPO to LLaVA-v1.5-7B and LLaVA-v1.5-13B across various hallucination benchmarks are presented in Tab. 1. The main findings are summarized as follows: (1) mPEA-DPO significantly reduces the hallucinations of the 7B and 13B models. Compared with the 7B (13B) base model, mPEA-DPO reduces the hallucination rate in MMHalBench by 33%, the response-level and object-level hallucination rates in Object HalBench by 92% and 87% respectively, and the CHAIR, hallucination rate, and human cognition in AMBER by 76%, 72%, and 86% respectively. (2) Compared with existing MLLMs alignment methods, for LLaVA-v1.5-13B, mPEA-DPO achieves the best performance on 75.0% of the hallucination metrics, and for LLaVA-v1.5-7B, it increases to 87.5%. (3) However, these enhancements lead to a slight compromise in coverage-related metrics. This indicates that models trained with mPEA-DPO tend to

adopt a slightly conservative strategy, avoiding uncertain assertions. Such a strategy enhances the credibility of the responses but may overlook some ambiguous details, which requires a trade-off.

## 5.3 IMPACT OF DATA SCALE

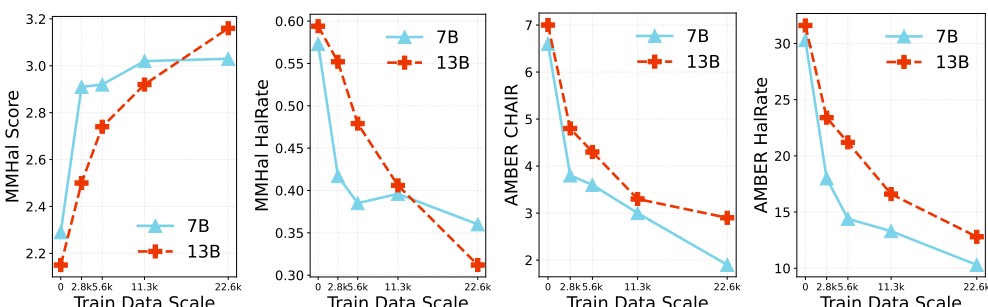

Figure 4: **Impact of data scale** on the performance of mPEA-DPO, using LLaVA as the base model. We assess: (1) the overall score and hallucination rate on MMHalBench, and (2) CHAIR and hallucination on AMBER.

To investigate the impact of data scale on mPEA-DPO, we present its performance under varying amounts of training data as Fig. 4. Even with only 2800 training instances, mPEA-DPO remains effective and outperforms the baseline. Additionally, we observe that the performance of mPEA-DPO consistently improves with increasing data scale, demonstrating significant performance gain.

## 5.4 ABLATION

**Impact of Component Combination.** To evaluate the contribution of each component in mPEA-DPO and the effect of their combinations, we conducted a comprehensive ablation study on mPEA-DPO based on LLaVA. The experimental results are shown in Tab. 2. The main observations are as follows: (1) Both Response quality Preference Optimization (mRPO) and Visual context Preference Optimization (mVPO) are effective. On the MMHalBench and AMBER datasets, both mRPO (mPEA-DPO-$\mathcal{L}_{mRPO}$) and mVPO

Table 2: The ablation results of mPEA-DPO based on LLaVA. Values in **bold** denote the best performance.

| Model | MMHalBench | | AMBER | |
|---|---|---|---|---|
| | Score↑ | Hal.↓ | C.↓ | Cog.↓ |
| LLaVA-v1.5-7B | 2.11 | 0.54 | 7.8 | 4.2 |
| mPEA-DPO | **3.02** | **0.36** | **1.9** | **0.6** |
| -$\mathcal{L}_{mRPO}$ | 2.44 | 0.45 | 4.2 | 0.9 |
| -$\mathcal{L}_{mVPO}$ | 2.81 | 0.38 | 2.5 | 0.9 |

(mPEA-DPO-$\mathcal{L}_{mVPO}$) outperform base model. This suggests that: (a) during optimization process, Response quality Preference Optimization (RPO) enables the model to concentrate on more challenging data points and mitigates overfitting in DPO; (b) the introduction of Visual context Preference Optimization (mVPO) enhances the model's alignment between the image and text. (2) The combination of visual context preference optimization and response quality preference optimization strategies makes preference optimization the most powerful.

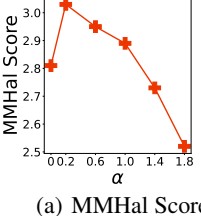 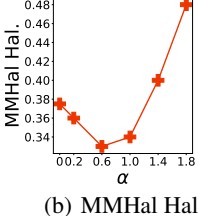 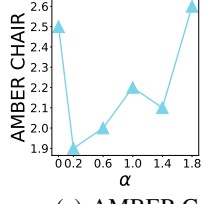 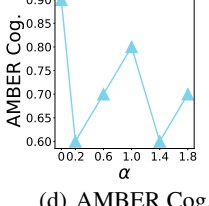

(a) MMHal Score  (b) MMHal Hal.  (c) AMBER C.  (d) AMBER Cog.

Figure 5: Results of mPEA-DPO evaluated on the MMHalBench and AMBER dataset with different choices of weight $\alpha$ to control the strength of Visual context Preference Optimization. Findings: when $\alpha = 0.2$, the best performance of the CHAIR and Hallucination Rate metric is achieved on AMBER based on LLaVA.

**Impact of Visual context Preference Optimization.** Visual context Preference Optimization (mVPO) forces models to make preference judgements based on critical visual context. Here, we discuss the impact of its weight. We fully consider Response quality Preference Optimization (mRPO) since its global textual semantics by setting its parameter to 1 in Eq. 17. As for Visual context Preference Optimization, given its crucial role in enhancing MLLM's attention to critical visual context, we fully explore the range of its weight $\alpha$ (as shown in Eq. 17). For the results of Fig. 5, we observe that the best performance was achieved when $\alpha = 0.2$ for LLaVA frameworks.

## 5.5 IMPACT OF REJECTED IMAGE CONSTRUCTION STRATEGY

The quality of visual preference data depends on the rejected image quality and its gap from the chosen images. In this section, we compare the impact of our proposed strategy for constructing rejected images and various existing strategies on preference optimization.

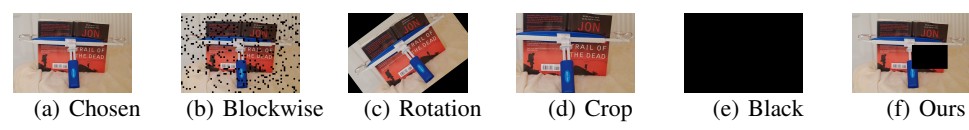

(a) Chosen    (b) Blockwise    (c) Rotation    (d) Crop    (e) Black    (f) Ours

Figure 6: Examples of rejected images constructed by different strategies. (a) is the chosen image.

**Strategies.** The existing rejected image construction strategies are listed below: (1) **Blockwise**: The chosen image is divided into blocks, with 30% of blocks randomly masked. (2) **Rotation**: The chosen image is randomly rotated between 10 to 80 degrees. (3) **Crop**: Random cropping strategy is applied to the chosen image. (4) **Blackness**: All RGB values in the chosen image are set to 0.

**Results.** The experimental results of mPEA-DPO under different construction strategies of rejected images are shown in Tab. 3. We observe **the rejected with removed key visual context can lead to better optimization results**. The blockwise, rotation, and crop strategies retain a significant amount of the chosen image's visual context, failing to effectively remove key visual context and thus leading to poorer performance. The blackness strategy, by completely masking the chosen image, virtually eliminates information about the chosen

Table 3: The ablation results of mPEA-DPO based on LLaVA. Values in **bold** denote the best performance.

| Strategy | MMHal Bench | | AMBER | |
|---|---|---|---|---|
| | Score↑ | Hal.↓ | CHAIR↓ | HalRate↓ |
| Ours | **3.03** | **0.36** | **1.9** | **10.3** |
| Blockwise | 2.64 | 0.42 | 2.8 | 14.1 |
| Rotation | 2.47 | 0.46 | 2.6 | 12.5 |
| Crop | 2.69 | 0.42 | 2.1 | 11.3 |
| Blackness | 2.90 | 0.41 | 2.4 | 12.3 |

image, resulting in poorer performance. However, Perception-Enhanced Preference Data enhances the sensitivity of MLLMs to key visual information by removing the key visual context from the chosen images, thereby achieving the best performance.

## 6 CONCLUSION

In summary, our study uncovers two fundamental issues of Direct Preference Optimization (DPO) in multimodal settings: (1) **Across-Image Insensitivity** and (2) **Within-Image Insensitivity**. Through both theoretical analysis and empirical evaluation, we systematically characterize the inherent limitations of existing multimodal DPO methods in exhibiting **visual insensitivity**. To address these limitations, we propose Perception-Enhanced Alignment (PEA)-DPO, a framework for MLLM alignment that explicitly leverages visual preference signals in conjunction with standard multimodal DPO. Experiments on three widely-used benchmarks demonstrate that PEA-DPO substantially improves the performance of LLaVA-v1.5-7B and LLaVA-v1.5-13B, achieving state-of-the-art results and surpassing other RLHF/RLAIF-based methods.

## ETHIC STATEMENT

Our work focuses on improving multimodal large language model (MLLM) alignment through preference-based optimization. All datasets used in this study, including AMBER and other publicly available benchmarks, were obtained under their respective licenses and ethical guidelines. We do not collect any private or sensitive user data. The proposed methods aim to reduce hallucinations and improve model reliability, thereby promoting safer and more trustworthy AI systems.

## REPRODUCIBILITY STATEMENT

We provide detailed descriptions of our methods, model architectures, training procedures, and hyperparameters in the main text and supplementary materials. All datasets are publicly available, and code necessary to reproduce our experiments will be released upon publication. Collectively, these resources are provided to enable precise reproduction and independent verification of our results.

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

## A  THE USE OF LARGE LANGUAGE MODELS (LLMS)

Note on Writing Assistance. Some sections of this paper have been lightly polished using large language models (LLMs) to improve clarity and readability. All scientific content, analyses, and interpretations remain the original work of the authors.

## B  PROOF

### B.1  PROOF OF THEOREM 1

**Theorem 1 (Across-Image Insensitivity)** *Suppose there exist samples* $(x, m_w, m_l, y_w, y_l)$ *such that*

$$M_\theta^y(x, m_w) - M_\theta^y(x, m_l) \leq \delta \quad (\delta \to 0), \tag{7}$$

*it implies the relation:*

$$\Delta_{\text{vis}}^\theta(x, m_w) - \Delta_{\text{vis}}^\theta(x, m_l) \leq \delta \quad (\delta \to 0). \tag{8}$$

**Proof 1** *Applying Bayes'rule, we have:*

$$\pi_\theta(y \mid x, m) = \frac{p_\theta(m \mid x, y)\,\pi_\theta(y \mid x)}{p_\theta(m \mid x)}. \tag{18}$$

*Taking the logarithm of both sides yields:*

$$\begin{aligned}
\log \pi_\theta(y \mid x, m) &= \log \pi_\theta(m \mid x, y) + \log \pi_\theta(y \mid x) - \log \pi_\theta(m \mid x) \\
&= \log \pi_\theta(y \mid x) + \ell_\theta(x, m; y).
\end{aligned} \tag{19}$$

*Substituting this into Eq. 5 gives:*

$$M_\theta^y(x, m) = \underbrace{\left[\log \pi_\theta(y_w \mid x) - \log \pi_\theta(y_l \mid x)\right]}_{\Delta_{\text{text}}^\theta(x)} + \underbrace{\left[\ell_\theta(x, m; y_w) - \ell_\theta(x, m; y_l)\right]}_{\Delta_{\text{vis}}^\theta(x,m)}. \tag{20}$$

*For the images* $m_w$ *and* $m_l$*, we have:*

$$M_\theta^y(x, m_w) = \Delta_{\text{text}}^\theta(x) + \Delta_{\text{vis}}^\theta(x, m_w), \tag{21}$$

$$M_\theta^y(x, m_l) = \Delta_{\text{text}}^\theta(x) + \Delta_{\text{vis}}^\theta(x, m_l), \tag{22}$$

*Taking the difference of Eq. 21 and Eq. 22 eliminates* $\Delta_{\text{text}}^\theta(x)$*, yielding:*

$$M_\theta^y(x, m_w) - M_\theta^y(x, m_l) = \Delta_{\text{vis}}^\theta(x, m_w) - \Delta_{\text{vis}}^\theta(x, m_l). \tag{23}$$

*When the empirical phenomenon in Eq. 7 is observed, it follows that:*

$$\Delta_{\text{vis}}^\theta(x, m_w) - \Delta_{\text{vis}}^\theta(x, m_l) \leq \delta \quad (\delta \to 0), \tag{24}$$

*which completes the proof.*

### B.2  PROOF OF THEOREM 2

**Theorem 2 (Within-Image Insensitivity)** *Suppose there exist samples* $(x, m_l, y)$ *such that*

$$\left|\ell_\theta(x, m_l; y)\right| \leq \varepsilon \quad (\varepsilon \to 0), \tag{9}$$

*and that the model exhibits **Across-Image Insensitivity**. Then the following holds for* $m_w$*:*

$$\left|\ell_\theta(x, m_w; y_w) - \ell_\theta(x, m_w; y_l)\right| \leq \delta + 2\varepsilon \quad (\delta, \varepsilon \to 0). \tag{10}$$

**Proof 2** *Substituting Eq. 9 and Eq. 5 into Eq. 7 and expanding, we obtain:*

$$\begin{aligned}
\left|\ell_\theta(x, m_w; y_w) - \ell_\theta(x, m_w; y_l)\right| &\leq \delta + \left|\ell_\theta(x, m_l; y_w)\right| + \left|\ell_\theta(x, m_l; y_l)\right| \\
&\leq \delta + 2\varepsilon \quad (\delta, \varepsilon \to 0),
\end{aligned} \tag{25}$$

*which completes the proof.*

## C  HOW DOES MPEA-DPO MITIGATE ACROSS- AND WITHIN-IMAGE INSENSITIVITY?

### C.1  THEORETICAL ANALYSIS

Given a policy $\pi_\theta$, assume the dataset contains sample $(x, m_w, m_l, y_w, y_l)$. According to the definition of $\Delta_{\mathrm{vis}}^\theta$ in Eq. 6, we have:

$$\Delta_{\mathrm{vis}}^\theta(x, m_w) = \ell_\theta(x, m_w; y_w) - \ell_\theta(x, m_w; y_l) \tag{26}$$

$$\Delta_{\mathrm{vis}}^\theta(x, m_l) = \ell_\theta(x, m_l; y_w) - \ell_\theta(x, m_l; y_l) \tag{27}$$

Then, based on the definition of the image-based likelihood ratio $\ell_\theta(x, m, y)$ in Eq. 4, we further obtain:

$$\Delta_{\mathrm{vis}}^\theta(x, m_w) = \log \frac{\pi_\theta(y_w \mid x, m_w)}{\pi_\theta(y_w \mid x)} - \log \frac{\pi_\theta(y_l \mid x, m_w)}{\pi_\theta(y_l \mid x)} \tag{28}$$

$$\Delta_{\mathrm{vis}}^\theta(x, m_l) = \log \frac{\pi_\theta(y_w \mid x, m_l)}{\pi_\theta(y_w \mid x)} - \log \frac{\pi_\theta(y_l \mid x, m_l)}{\pi_\theta(y_l \mid x)} \tag{29}$$

We subtract $\Delta_{\mathrm{vis}}^\theta(x, m_l)$ from $\Delta_{\mathrm{vis}}^\theta(x, m_w)$ to obtain:

$$\begin{aligned}
\Delta_{\mathrm{vis}}^\theta(x, m_w) - \Delta_{\mathrm{vis}}^\theta(x, m_l) =& \log \frac{\pi_\theta(y_w \mid x, m_w)}{\pi_\theta(y_w \mid x)} - \log \frac{\pi_\theta(y_w \mid x, m_l)}{\pi_\theta(y_w \mid x)} \\
&+ \log \frac{\pi_\theta(y_l \mid x, m_l)}{\pi_\theta(y_l \mid x)} - \log \frac{\pi_\theta(y_l \mid x, m_w)}{\pi_\theta(y_l \mid x)} \\
=& \log \pi_\theta(y_w \mid x, m_w) - \log \pi_\theta(y_w \mid x, m_l) \\
&+ \log \pi_\theta(y_l \mid x, m_l) - \log \pi_\theta(y_l \mid x, m_w)
\end{aligned} \tag{30}$$

When the policy $\pi_\theta$ exhibits **Across-Image Insensitivity** (*i.e.,* **Theorem 1**) on this sample, there exists a constant $\delta \to 0$ such that:

$$\begin{aligned}
&\Delta_{\mathrm{vis}}^\theta(x, m_w) - \Delta_{\mathrm{vis}}^\theta(x, m_l) = \\
&\log \pi_\theta(y_w \mid x, m_w) - \log \pi_\theta(y_w \mid x, m_l) + \log \pi_\theta(y_l \mid x, m_l) - \log \pi_\theta(y_l \mid x, m_w) \leq \delta
\end{aligned} \tag{31}$$

Thus, we obtain the formulation of **Across-Image Insensitivity** in terms of log-probabilities:

$$\underbrace{\log \pi_\theta(y_w \mid x, m_w) - \log \pi_\theta(y_w \mid x, m_l)}_{\text{first term}} + \underbrace{\log \pi_\theta(y_l \mid x, m_l) - \log \pi_\theta(y_l \mid x, m_w)}_{\text{second term}} \leq \delta \tag{32}$$

The inequality in Eq. 32 decomposes into two terms, each reflecting how the policy $\pi_\theta$ responds when the conditioning image is changed **across** distinct visual inputs. The first term measures how much the correct visual evidence $m_w$ increases the model's confidence in the preferred response $y_w$. It captures the degree to which the model positively leverages the image to support the correct output. A large value means that the model meaningfully associates the preferred response with the correct image. The second term captures how much more likely the model is to produce the dispreferred output under the mismatched image $m_l$ compared to the correct image $m_w$.

In practical multimodal scenarios, improving the first term has a more direct and substantial impact on overall performance. Enhancing the model's preference for the preferred response under the correct image explicitly boosts the probability that the model produces the task-relevant, correct output. This directly affects answer quality and alignment with human-preferred behavior. In contrast, improving the second term primarily reduces the preference of undesired responses, which does not necessarily guarantee that the preferred response becomes the top candidate. Therefore, optimizing the first term, which corresponds to the increased likelihood of the preferred response, is generally more important for improving model correctness and practical task performance.

Similarly, based on the definition of the image-based likelihood ratio $\ell_\theta(x, m, y)$ in Eq. 4, we have:

$$\ell_\theta(x, m_w; y_w) = \frac{\log \pi_\theta(y_w \mid x, m_w)}{\log_\theta(y_w \mid x)} \tag{33}$$

$$\ell_\theta(x, m_w; y_l) = \frac{\log \pi_\theta(y_l \mid x, m_w)}{\log_\theta(y_l \mid x)} \tag{34}$$

Subtracting $\ell_\theta(x, m_w; y_l)$ from $\ell_\theta(x, m_w; y_w)$ yields:

$$\begin{aligned}
\ell_\theta(x, m_w; y_w) - \ell_\theta(x, m_w; y_l) =& \frac{\log \pi_\theta(y_w \mid x, m_w)}{\log_\theta(y_w \mid x)} - \frac{\log \pi_\theta(y_l \mid x, m_w)}{\log_\theta(y_l \mid x)} \\
=& \log \pi_\theta(y_w \mid x, m_w) - \log \pi_\theta(y_l \mid x, m_w) \\
&+ \log_\theta(y_l \mid x) - \log_\theta(y_w \mid x)
\end{aligned} \tag{35}$$

When the policy $\pi_\theta$ exhibits a **Within-Image Insensitivity** (*i.e.,* **Theorem 2**) on this sample, there exists a constant $\eta = \delta + 2\varepsilon \to 0$ such that:

$$\begin{aligned}
|\ell_\theta(x, m_w; y_w) - \ell_\theta(x, m_w; y_l)| = \\
|\log \pi_\theta(y_w \mid x, m_w) - \log \pi_\theta(y_l \mid x, m_w) + \log_\theta(y_l \mid x) - \log_\theta(y_w \mid x)| \le \eta
\end{aligned} \tag{36}$$

Thus, we obtain the formulation of **Within-Image Insensitivity** in terms of log-probabilities:

$$|\underbrace{\log \pi_\theta(y_w \mid x, m_w) - \log \pi_\theta(y_l \mid x, m_w)}_{\text{first term}} + \underbrace{\log_\theta(y_l \mid x) - \log_\theta(y_w \mid x)}_{\text{second term}}| \le \eta \tag{37}$$

Similarly, the inequality in Eq. 37 consists of two terms, each reflecting a distinct aspect of the model's behavior. The first term measures the model's ability to discriminate the preferred response $y_w$ rom the dispreferred response $y_l$ when conditioned on both the text prompt $x$ and the image $m_w$. It captures how effectively the model leverages the visual input to favor semantically relevant outputs over irrelevant ones. A larger value indicates stronger visual grounding and better within-image discrimination. The second term captures the difference between dispreferred and preferred responses without any image input, reflecting the model's inherent text-only biases.

In practical multimodal scenarios, optimizing the first term is generally more critical because it directly enhances the model's ability to leverage the image to generate the preferred response. Increasing this term ensures that the correct response is more likely when the image is provided, which has important impact on task performance and effective visual grounding. By contrast, adjusting the second term only influences text-only biases. It does not guarantee that the preferred response will dominate when the image is present. Therefore, prioritizing the first term is essential for improving correctness and within-image sensitivity.

Let $G_m = \log \pi_\theta(y_w \mid x, m_w) - \pi_\theta(y_w \mid x, m_l)$, $G_r = \log \pi_\theta(y_w \mid x, m_w) - \log \pi_\theta(y_l \mid x, m_w)$. Thus, our goal is to increase $G_m$ to mitigate **Across-Image Insensitivity**, and to increase $G_r$ to mitigate **Within-Image Insensitivity**.

Revisiting the loss function of mPEA-DPO, we have:

$$\begin{aligned}
\mathcal{L}_{\text{mPEA-DPO}} =& \mathcal{L}_{\text{mRPO}} + \alpha \cdot \mathcal{L}_{\text{mVPO}} \\
=& \mathbb{E}_\mathcal{D} \left\{ \text{ReLU} \left[ -\left( \frac{\log \pi_\theta(y_w \mid x, m_w)}{|y_w|} - \frac{\log \pi_\theta(y_l \mid x, m_w)}{|y_l|} - \gamma_r \right) \right] \right\} \\
&+ \alpha \cdot \mathbb{E}_{\mathcal{D}_m} \left\{ \text{ReLU} \left[ -\left( \frac{\log \pi_\theta(y_w \mid x, m_w)}{|y_w|} - \frac{\log \pi_\theta(y_w \mid x, m_l)}{|y_w|} - \gamma_m \right) \right] \right\},
\end{aligned} \tag{38}$$

As shown in Eq. 38, the proposed mPEA-DPO consists of two components: $\mathcal{L}_{\text{mRPO}}$, which mitigates **Within-Image Insensitivity**, and $\mathcal{L}_{\text{mVPO}}$, which mitigates **Across-Image Insensitivity**. $\mathcal{L}_{\text{mPEA-DPO}}$ is a linear combination of $\mathcal{L}_{\text{mRPO}}$ and $\mathcal{L}_{\text{mVPO}}$, effectively addressing both Across- and Within-Image Insensitivity observed in MLLMs under real-world scenarios.

Notably, inspired by SimPO (Meng et al., 2024), we normalize the log-probabilities in the mPEA-DPO loss function instead of using log-probabilities from $G_m$ and $G_r$. This normalization is applied to prevent length bias in the model outputs.

## C.2 EXPERIMENTS

**Experimental Setup.** To evaluate whether mPEA-DPO can effectively mitigate both **Across-Image Insensitivity** and **Within-Image Insensitivity** by jointly enlarging $G_m$ and $G_r$, we conduct a controlled validation experiment built upon the LLaVA. We use 25% of the training data to train two models, one optimized with mPEA-DPO objective $\mathcal{L}_{\text{mPEA-DPO}}$ amd the other with the standard DPO objective $\mathcal{L}_{\text{DPO}}$. The remaining 75% of the data is used as a test set to estimate the empirical distributions of $G_m$ and $G_r$.

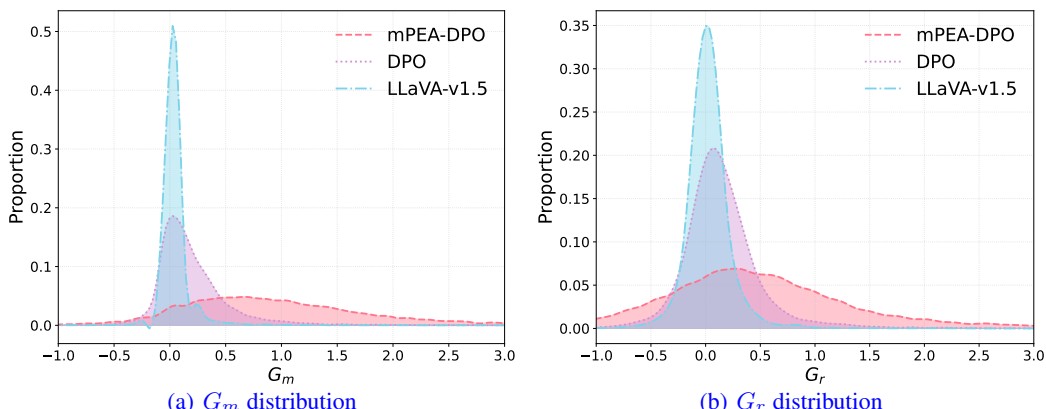

(a) $G_m$ distribution          (b) $G_r$ distribution

Figure 7: Comparison of $G_m$ and $G_r$ distribution.

**Results.** The experimental results are presented in Fig. 7. The main findings are summarized as follows: (1) Under the DPO objective, the distribution of $G_m$ is heavily concentrated around zero. This indicates that the model fails to distinguish chosen images from rejected images, which reflects strong **Across-Image Insensitivity**. (2) The mPEA-DPO objective substantially increases both $G_m$ and $G_r$. This demonstrates that the method effectively eliminates **Across-Image Insensitivity** and **Within-Image Insensitivity** simultaneously.

## D  ANALYSIS IN CONSTRUCTION OF PERCEPTION-ENHANCED PREFERENCE DATA

To provide a systematic analysis of constructing perception-enhanced preference data, we conduct a comprehensive study based on LLaVA. Specifically, we investigate how different design choices affect the quality of perception-enhanced preference data. Our analysis focuses on three key aspects: (1) the number of candidate masks, (2) the masking ratio, and (3) the similarity metrics.

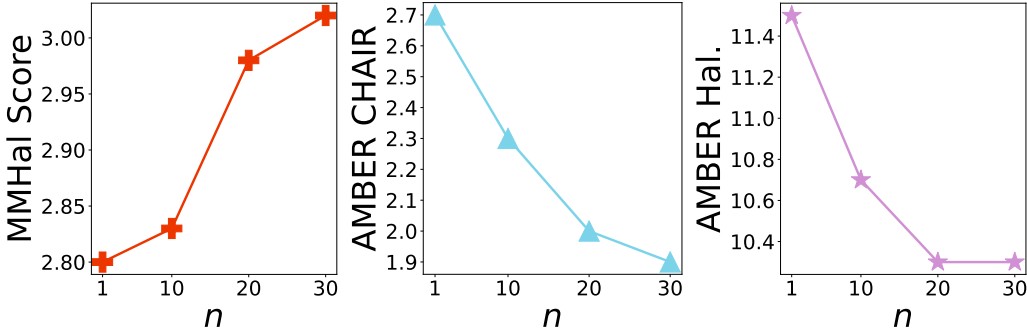

Figure 8: **Impact of the number of candidate masks** on the quality of perception-enhanced preference data, using CLIP embedding. We assess: (1) the overall score on MMHalBench, and (2) CHAIR and hallucination on AMBER.

**The impact of the number of candidate masks.** As shown in Fig. D, we report the performance of mPEA-DPO under varying values of the number of candidate masks $n$, and begins to plateau once $n \geq 20$. This indicates that a larger set of candidate masks tends to yield higher-quality rejected images, though at the cost of increased computation. Thus, selecting $n$ involves a trade-off between performance and computational overhead.

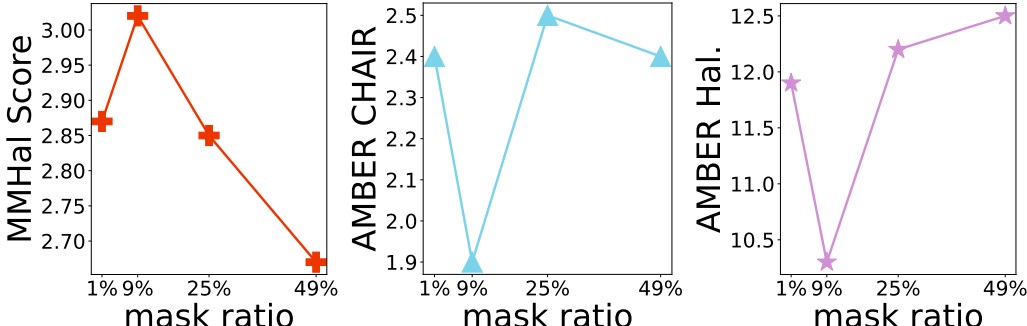

Figure 9: **Impact of the masking ratio** on the quality of perception-enhanced preference data, using CLIP embedding. We assess: (1) the overall score on MMHalBench, and (2) CHAIR and hallucination on AMBER.

**The impact of the masking ratio.** As shown in Fig. D, we evaluate the performance of mPEA-DPO under different masking ratios. We observe that mPEA-DPO achieves its best performance when the masking ratio is around 9%. In contrast, using either too small or too large a masking ratio leads to degraded performance. This suggests that removing the key visual context of the chosen image is crucial for constructing effective rejected images and provides the best performance gain.

Table 4: **The impact of similarity metrics** on the quality of perception-enhanced preference data, using CLIP embedding. We assess: (1) the overall score and hallucination on MMHalBench, and (2) CHAIR and hallucination on ABMER.

|  | MMHalBench | | AMBER | | | |
| --- | --- | --- | --- | --- | --- | --- |
|  | Score ↑ | Hal. ↓ | C. ↓ | Cover. ↑ | Hal. ↓ | Cog. ↓ |
| Euclidean Distance | 2.72 | 0.35 | 2.3 | 47.4 | 11.4 | 0.6 |
| Cosine Similarity | 3.02 | 0.36 | 1.9 | 46.7 | 10.3 | 0.6 |

**The impact of similarity metrics** To investigate how different similarity metrics influence the construction of perception-enhanced preference pairs, we compare two commonly used measures, Euclidean distance and cosine similarity, both computed using CLIP embeddings. As shown in Tab. 4, these two metrics lead to different performance for mPEA-DPO. We observe that cosine similarity consistently outperforms Euclidean distance. A plausible explanation is that CLIP (Radford et al., 2021) is pretrained with cosine similarity as its alignment objective, which allows it to better exploit cosine similarity during inference when evaluating the similarity between images.

## E GENERALIZATION OF MPEA-DPO ACROSS DIFFERENT VISUAL ENCODERS

Recent studies (Tong et al., 2024) suggest that different visual encoders exhibit different "blind spots". To verify that our method can effectively construct perception-enhanced preference data across different visual encoders, we conduct experiments comparing two widely used encoders, DINO (Caron et al., 2021) and CLIP (Radford et al., 2021), and evaluate their impact on the performance of mPEA-DPO. The results are presented in Tab. 5.

Table 5: **The impact of visual encoder type** on the quality of perception-enhanced preference data. We assess: (1) the overall score and hallucination on MMHalBench, (2) CHAIRs and CHAIRi on Object HalBench, and (3) CHAIR and hallucination on ABMER.

| | MMHalBench | | Object HalBench | | AMBER | | | |
|---|---|---|---|---|---|---|---|---|
| | Score ↑ | Hal. ↓ | $CHAIR_s$ ↓ | $CHAIR_i$ ↓ | C. ↓ | Cover. ↑ | Hal. ↓ | Cog. ↓ |
| mPEA-DPO (w/ DINO) | 2.96 | 0.35 | 4.7 | 3.0 | 2.2 | 46.6 | 10.5 | 0.7 |
| mPEA-DPO (w/ CLIP) | 3.02 | 0.36 | 4.3 | 3.2 | 1.9 | 46.7 | 10.3 | 0.6 |

We observe that both the CLIP-based and DINO-based construction strategies effectively reduce hallucination across all benchmarks. This demonstrates that our approach is not tied to a specific encoder and generalizes well to different types of visual representations.

# F  ATTRIBUTION OF REDUCED HALLUCINATIONS

As discussed in (Amirloo et al., 2024), the CHAIR metric has a known limitation in that it does not penalize shorter responses, which may trivially reduce hallucination scores. To demonstrate that the reduction in CHAIR metrics is indeed driven by reduced hallucination rather than shorter or less informative outputs, we conducted a comparison on Object HalBench between our method and two of the strongest existing baselines, OPA-DPO (Yang et al., 2025) and DAMA (Lu et al., 2025). We report three metrics: CHAIRs, CHAIRi, and Recall.

Table 6: Comparison of hallucination and recall performance on Object HalBench. Values in **bold** denote the best performance.

| | CHAIRs | CHAIRi | Recall |
|---|---|---|---|
| OPA-DPO | 13.3 | 4.3 | 43.29 |
| DAMA | 10.3 | 5.9 | **54.50** |
| mPEA-DPO | **4.3** | **3.2** | 52.3 |

As shown in Tab. 6, mPEA-DPO achieves clearly superior hallucination metrics compared with both OPA-DPO and DAMA. Importantly, mPEA-DPO attains recall performance comparable to DAMA while substantially outperforming OPA-DPO, indicating that the improvement in CHAIR metrics is not simply due to shorter responses but rather reflects a genuine reduction in hallucination.

# G  IMPACT OF MVPO ON GENERAL CAPABILITIES

To evaluate whether mVPO affects general capabilities, we conducted experiments comparing mPEA-DPO with and without the mVPO component on two general capabilities benchmarks, MMStar (Chen et al., 2024) and AI2D (Kembhavi et al., 2016). The results are shown in Tab. 7. We observe that removing mVPO leads to only a slight decrease in performance on both benchmarks, indicating that including mVPO does not harm general capabilities and in fact enhances the model's visual understanding.

Table 7: Results of mPEA-DPO and mPEA-DPO without mVPO on MMStar and AI2D based on LLaVA-v1.5 . Values in **bold** denote the best performance.

| | MMStar | AI2D |
|---|---|---|
| mPEA-DPO | **3.02** | **0.36** |
| -mVPO | 2.81 | 0.38 |

Furthermore, we provide a detailed results across six core capabilities in MMStar. These results in Tab. 8 further confirm that our method does not degrade general capabilities.

Table 8: A detailed results of mPEA-DPO and mPEA-DPO without mVPO across the six core capabilities in MMStar: CP(coarse perception), FP(fine-grained perception), IR(instance reasoning), LR(logical reasoning), ST(science & technology), MA(mathematics).Values in **bold** denote the best performance.

|          | CP       | FP       | IR       | LR       | SR       | MA       | AVG      |
|----------|----------|----------|----------|----------|----------|----------|----------|
| mPEA-DPO | **62.4** | **30.0** | 39.6     | 23.6     | **24.0** | 17.2     | **32.8** |
| -mVPO    | 60.8     | 28.8     | **41.2** | **24.4** | 23.2     | **17.6** | 32.6     |

## H   FAIR COMPARISON WITH COMPETITIVE BASELINES

The core components of our proposed method are: (1) the construction of rejected images and (2) a novel optimization objective. To ensure a fair comparison, we evaluated our method using the same base model, training datasets, and evaluation settings as the baselines. We selected two competitive methods, mDPO (Wang et al., 2024) and DAMA (Lu et al., 2025), and compared them with our approach across three benchmarks: MMHal-Bench, Object HalBench, and AMBER. The experimental results are summarized in the Tab. 9.

Table 9: A fair comparison of mPEA-DPO against competitive baselines. We assess: (1) the overall score and hallucination on MMHalBench, (2) CHAIRs and CHAIRi on Object HalBench, and (3) CHAIR and hallucination on ABMER.

|                  | MMHalBench | | Object HalBench | | AMBER | | | |
|------------------|-----------|---------|-----------------------|-----------------------|---------|------------|---------|---------|
|                  | Score ↑   | Hal. ↓  | CHAIR$_s$ ↓ | CHAIR$_i$ ↓ | C. ↓    | Cover. ↑   | Hal. ↓  | Cog. ↓  |
| mDPO             | 2.39      | 0.52    | 39.3                  | 20.4                  | 4.7     | 49.5       | 21.4    | 2.2     |
| DAMA             | 2.76      | 0.41    | 10.3                  | 5.9                   | 3.0     | 48.3       | 14.8    | 1.2     |
| mPEA-DPO (Ours)  | 3.02      | 0.36    | 4.3                   | 3.2                   | 1.9     | 46.7       | 10.3    | 0.6     |

From these results, we draw the following conclusion: under identical base model, training and evaluation settings, mPEA-DPO demonstrates superior hallucination reduction compared to mDPO and DAMA, validating the strong performance of our method.

## I   GENERAL CAPABILITY ANALYSIS

Preference optimization may negatively affect the model's generalization ability. In this section, we evaluate and analyze the general capabilities of MLLM enhanced with our mPEA-DPO. Specifically, we consider several widely used general capability benchmarks, including MMStar (Chen et al., 2024), AI2D (Kembhavi et al., 2016) and MMMU (Yue et al., 2024a). We compare the performance of LLaVA and LLaVA+mPEA-DPO on datasets. The results are presented in Tab. 10.

Table 10:  The general capability evaluation results. Values in **bold** denote the best performance.

|               | MMStar   | AI2D     | MMMU     |
|---------------|----------|----------|----------|
| LLaVA-v1.5-7B | 30.3     | 49.1     | **35.3** |
| +mPEA-DPO     | **32.8** | **51.9** | 33.4     |

We observe that LLaVA+mPEA-DPO outperforms LLaVA on two out of the three datasets. This indicates that the mPEA-DPO-enhanced LLaVA achieves slight improvements on the MMStar and AI2D datasets, while maintaining comparable performance on the MMMU dataset.

Furthermore, as shown in Tab. 11, we provide a breakdown of results across the six core capabilities in MMStar, with each category containing 250 samples. We observe that LLaVA+mPEA-DPO outperforms LLaVA on four out of six capabilities, while achieving comparable performance on one additional capability. These results further demonstrate that our method does not compromise the model's general capabilities.

Table 11: A breakdown of results across the six core capabilities in MMStar: CP(coarse perception), FP(fine-grained perception), IR(instance reasoning), LR(logical reasoning), ST(science & technology), MA(mathematics).Values in **bold** denote the best performance.

|  | CP | FP | IR | LR | SR | MA | AVG |
|---|---|---|---|---|---|---|---|
| LLaVA-v1.5-7B | 58.8 | 24.0 | 38.8 | **24.0** | 13.6 | **22.8** | 30.3 |
| +mPEA-DPO | **62.4** | **30.0** | **39.6** | 23.6 | **24.0** | 17.2 | **32.8** |

Additionally, we provide a comprehensive comparison of the performance of LLaVA and LLaVA+mPEA-DPO on the MMMU dataset. Fig. 10 show the overall accuracy across 36 major disciplinary domains. mPEA-DPO largely maintains the performance of LLaVA, with only a negligible decrease. We hypothesize that this slight drop arises because mPEA-DPO adopts a conservative strategy when confronted with particularly challenging cases, thereby avoiding uncertain or potentially incorrect assertions.

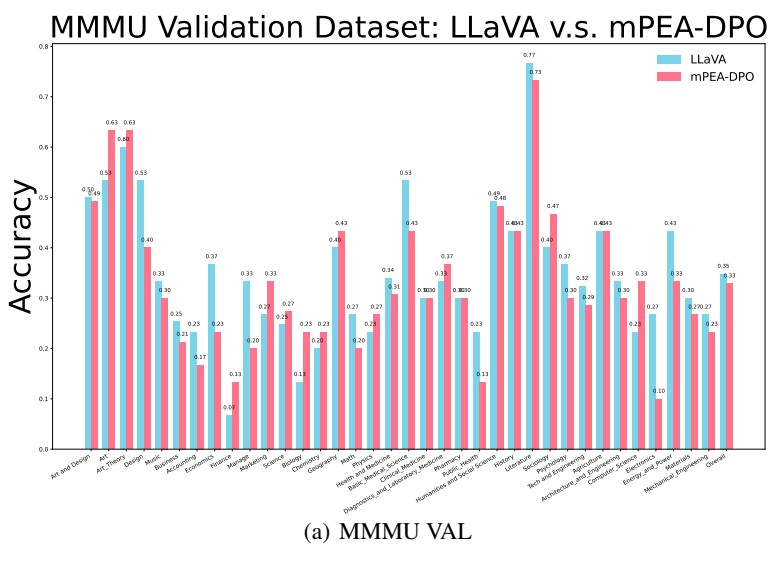

(a) MMMU VAL

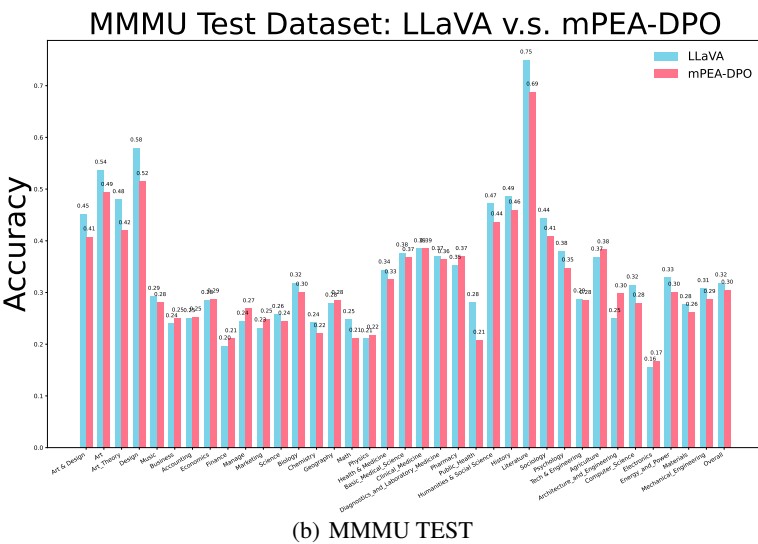

(b) MMMU TEST

Figure 10: Overall accuracy comparison between LLaVA and mPEA-DPO on (a) the MMMU validation set and (b) the MMMU test set, reported across 36 major disciplinary domains, along with the global average. The validation set contains 900 samples, while the test set contains 10,500 samples.

