# OpenReview forum: "PEA-DPO: Perception-Enhanced Alignment Direct Preference Optimization for MLLMs Alignment"
_ICLR.cc/2026/Conference — Submitted to ICLR 2026_

### Official Review · Reviewer_KtuX · 2025-10-20

**Soundness:** 3
**Presentation:** 3
**Contribution:** 3
**Rating:** 6
**Confidence:** 3

**Summary:**

This paper identifies a key failure mode in multimodal DPO: visual insensitivity, with across-image and within-image manifestations. The authors propose PEA-DPO, a dual-preference objective that optimizes both response-quality preferences and visual-context preferences. The visual-context signal is built by creating masked variants of the original image, selecting the most semantics-damaging one via CLIP similarity, and encouraging the model to prefer responses conditioned on the full image. A reference-free, margin-based ReLU variant filters trivial pairs and reduces compute. On LLaVA-1.5 with 7B and 13B backbones, the method substantially reduces hallucinations and delivers strong results on MMHal Bench, Object HalBench, and AMBER, with a small tradeoff in coverage and a slight regression on MMMU validation and test.

**Strengths:**

1. The paper clearly defines visual insensitivity, decomposes preferences into textual and visual components, and articulates both across-image and within-image failure modes.
2. The reference-free ReLU-margin formulation removes dependence on a reference model, filters trivial pairs, and improves training efficiency and robustness.
3. The experiments include clear ablations of components and the weighting hyperparameter,  along with scaling results that support the design choices and show effectiveness even with modest amounts of training data.
4. The method achieves large and consistent reductions in multimodal hallucinations on MMHal-Bench, Object HalBench, and AMBER for both 7B and 13B LLaVA backbones, while largely maintaining general capability with a small coverage tradeoff and a slight regression on MMMU.

**Weaknesses:**

1. The masking strategy relies on several design choices, including the masking ratio, the number of candidate masks, and whether to use CLIP cosine similarity. However, the final selections are not supported by a systematic study.
2. The comparability to some strong proprietary systems and newer alignment methods is imperfect due to differences in base models, data, and evaluation settings, which weakens the strength of SOTA claims.
3. The CLIP-guided selection of the “most damaging” mask requires generating and scoring many candidates, which introduces non-trivial data-generation overhead and complicates scaling.

**Questions:**

1. Could VPO degrade language-only capabilities or long-context reasoning? Any broader evals beyond hallucination (e.g., MMMU shown, but more detail would help).
2. How does PEA-DPO perform with other models? Please report results and adaptation details to evaluate generalization.
3. Please provide a detailed analysis of the MMMU regression to clarify whether the decline reflects increased conservativeness or a genuine loss of capability.
4. As a diagnostic follow up to item 3, please evaluate PEA-DPO on MMStar to test performance under genuinely vision-dependent settings and to reduce text-only shortcuts, and also report calibration or selectivity metrics such as risk and coverage curves, Accuracy at Coverage with an abstain option, and ECE or Brier scores to separate lucky guesses from grounded answers?

---

> ### Author Response · Authors · 2025-11-23
> **Response to Reviewer KtuX (1/3)**
>
> We sincerely thank you for your valuable feedback and thoughtful suggestions. Below, we provide detailed responses to the concerns and questions raised.
>
> ------------------------------
>
> **Weakness**
>
> **W1: The masking strategy relies on several design choices, including the masking ratio, the number of candidate masks, and whether to use CLIP cosine similarity. However, the final selections are not supported by a systematic study.**
>
> R1: We conducted comprehensive experiments to systematically study the effects of various hyperparameters, including the masking ratio, the number of candidate masks, and the CLIP cosine-similarity threshold. Detailed results are provided in ${\color{blue}\text{Appendix D}}$.
>
> **W2: The comparability to some strong proprietary systems and newer alignment methods is imperfect due to differences in base models, data, and evaluation settings, which weakens the strength of SOTA claims.**
>
> R2: The core components of our proposed method are: (1) the construction of rejected images and (2) a novel optimization objective.  To ensure a fair comparison, we evaluated our method using the same base model, training datasets, and evaluation settings as the baselines. We selected two competitive methods, mDPO[1] and DAMA[2], and compared them with our approach across three benchmarks: MMHal-Bench, Object HalBench, and AMBER. The experimental results are summarized in the table below.
>
> From these results, we draw the following conclusion: under identical base model, training and evaluation settings, mPEA-DPO demonstrates superior hallucination reduction compared to mDPO and DAMA, validating the strong performance of our method.
>
> * MMHal-Bench:
>
> |  | Score | Hal. |
> |-------|-------|-------|
> | mDPO | 2.39 | 0.52 |
> | DAMA | 2.76 | 0.41 |
> | mPEA-DPO | 3.02 | 0.36 |
>
> * Object HalBench:
>
> |  | CHAIRs | CHAIRi |
> |-------|-------|-------|
> | mDPO | 39.3 | 20.4 |
> | DAMA | 10.3 | 5.9 |
> | mPEA-DPO | 4.3 | 3.2 |
>
> * AMBER:
>
> |  | C. | Cover | Hal. | Cog |
> |-------|-------|-------|-------|-------|
> | mDPO | 4.7 | 49.5 | 21.4 | 2.2 |
> | DAMA | 3.0 | 48.3 | 14.8 | 1.2 |
> | mPEA-DPO | 1.9 | 46.7 | 10.3 | 0.6 |
>
> [1] Wang, Fei, et al., mdpo: Conditional preference optimization for multimodal large language models. EMNLP 2024.
>
> [2] Lu, Jinda, et al., DAMA: Data- and Model-aware Alignment of Multi-modal LLMs. ICML 2025.
>
> **W3: The CLIP-guided selection of the “most damaging” mask requires generating and scoring many candidates, which introduces non-trivial data-generation overhead and complicates scaling.**
>
> R3: During training, the construction of rejected images is fully decoupled from model optimization. This means that once constructed, the rejected images can be reused across training runs without additional cost. We measured the overhead of our CLIP-guided construction process on 22.6k samples with a candidate mask set size of 20, and the total time was 5 hours and 34 minutes.
>
> |  | Time |
> |-------|-------|
> | Data Construction | 5h34m |
>
> We also computed the training time for mDPO and our proposed mPEA-DPO. The results are shown below. Since our method removes the need for a reference model, the training time of mPEA-DPO is actually lower than that of mDPO, while achieving superior hallucination reduction performance.
>
> |  | Training Time |
> |-------|-------|
> | mDPO | 21h03m |
> | mPEA-DPO | 18h32m |

---

> ### Author Response · Authors · 2025-11-23
> **Response to Reviewer KtuX (2/3)**
>
> **Question**
>
> **Q1: Could VPO degrade language-only capabilities or long-context reasoning? Any broader evals beyond hallucination**
>
> A1: To evaluate whether VPO affects language-only capabilities or long-context reasoning, we conducted experiments comparing mPEA-DPO with and without the mVPO component on two quality-focused benchmarks, MMStar and AI2D. The results are summarized below. We observe that removing mVPO leads to only a slight decrease in performance on both benchmarks, indicating that including mVPO does not harm language-only capabilities and in fact enhances the model’s visual understanding.
>
> | | MMStar | AI2D |
> |-------|-------|-------|
> | mPEA-DPO (Ours) | 32.8 | 51.9 |
> | - mVPO | 32.6 | 50.8 |
>
> Furthermore, we provide a breakdown of results across six core capabilities in MMStar, with each category containing 250 samples. These results further confirm that our method does not degrade language-only capabilities or long-context reasoning. On the contrary, the model demonstrates improved overall performance after post-alignment.
>
> |  | CP | FP | IR | LR | ST | MA | AVG|
> |-------|-------|-------|-------|-------|-------|-------|-------|
> | mPEA-DPO (Ours) | 62.4 | 30.0 | 39.6 | 23.6 | 24.0 | 17.2 | 32.8 |
> | - mVPO | 60.8 | 28.8 | 41.2 | 24.4 | 23.2 | 17.6 | 32.6 |
>
>
> **Q2: How does PEA-DPO perform with other models? Please report results and adaptation details to evaluate generalization.**
>
> A2: We appreciate the reviewer’s interest in understanding the generalization ability of our method. We evaluate mPEA-DPO along two dimensions: generalization across visual encoders and generalization across base models. Here, we first report the results for visual encoders. Since full training on additional base models requires more time, here we report the results for visual encoders. We will make our best effort to include these results and analyses during the rebuttal period; otherwise, we will provide them in the final version.
>
> To assess the generalizability of our rejected image construction method, we used DINOv2-base as an alternative visual encoder to guide the construction of rejected images. We evaluated mPEA-DPO on MMHal-Bench, Object HalBench, and AMBER. The results are presented below.
>
> We observe that both the CLIP-based and DINO-based construction strategies effectively reduce hallucination across all benchmarks. This demonstrates that our approach is not tied to a specific encoder and generalizes well to different types of visual representations.
>
> * MMHal-Bench:
>
> |  | Score | Hal. |
> |-------|-------|-------|
> | mPEA-DPO (DINO) | 2.96 | 0.35 |
> | mPEA-DPO (CLIP)| 3.02 | 0.36 |
>
>
> * Object HalBench:
>
> |  | CHAIRs | CHAIRi |
> |-------|-------|-------|
> | mPEA-DPO (DINO)| 4.7 | 3.0 |
> | mPEA-DPO (CLIP) | 4.3 | 3.2 |
>
> * AMBER:
>
> |  | C. | Cover | Hal. | Cog |
> |-------|-------|-------|-------|-------|
> | mPEA-DPO (DINO) | 2.2 | 46.6 | 10.5 | 0.7 |
> | mPEA-DPO (CLIP) | 1.9 | 46.7 | 10.3 | 0.6 |

---

> ### Author Response · Authors · 2025-11-23
> **Response to Reviewer KtuX (3/3)**
>
> **Question**
>
> **Q3: Please provide a detailed analysis of the MMMU regression to clarify whether the decline reflects increased conservativeness or a genuine loss of capability.**
>
> A3: We sincerely thank the reviewer for this insightful question. In our manuscript, we hypothesized that the slight decrease in MMMU accuracy is due to the model adopting a more conservative behavior after training. Since MMMU is entirely composed of multiple-choice questions, it is difficult to determine from textual response patterns whether the observed decline reflects increased conservativeness or a genuine loss of capability.
>
> To further investigate, we measured both accuracy and recall on Object HalBench, which is an image captioning task. The results are shown below. We observe that mPEA-DPO achieves higher accuracy than the base model but exhibits lower recall, suggesting that the model tends to adopt a more conservative strategy after post-alignment.
>
> | | Acc | Recall |
> |-------|-------|-------|
> | LLaVA-v.15-7B | 73.6 | 64.5 |
> | mPEA-DPO (Ours) | 95.6 | 52.4 |
>
> **Q4: Please evaluate PEA-DPO on MMStar to test performance.**
>
> A4: We evaluated the overall performance of mPEA-DPO on the MMStar dataset, and the results are summarized in the table below. We observe that mPEA-DPO outperforms its backbone model LLaVA-v1.5-7B.
>
> | | MMStar |
> |-------|-------|
> | LLaVA-v.15-7B | 30.3 |
> | mPEA-DPO (Ours) | 32.8 |
>
> Furthermore, we provide a detailed breakdown of the six core capabilities in MMStar, with each category containing 250 samples. These results further demonstrate that our method does not degrade the model’s overall performance. On the contrary, mPEA-DPO improves the general capabilities of LLaVA-v1.5.
>
> CP (coarse perception), FP (fine-grained perception), IR(instance reasoning), LR (logical reasoning), ST (science & technology), MA (mathematics).
>
> |  | CP | FP | IR | LR | ST | MA | AVG|
> |-------|-------|-------|-------|-------|-------|-------|-------|
> | LLaVA-v.15-7B | 58.8 | 24.0 | 38.8 | 24.0 | 13.6 | 22.8 | 30.3 |
> | mPEA-DPO (Ours) | 62.4 | 30.0 | 39.6 | 23.6 | 24.0 | 17.2 | 32.8 |

---

### Official Review · Reviewer_kyXb · 2025-10-28

**Soundness:** 3
**Presentation:** 2
**Contribution:** 2
**Rating:** 4
**Confidence:** 4

**Summary:**

The paper investigates the issue of "visual insensitivity" in Multimodal Large Language Models (MLLMs) during Direct Preference Optimization (DPO) alignment. The authors point out that MLLMs suffer from two specific problems: across-image insensitivity and within-image insensitivity. Through theoretical analysis, they demonstrate that visual insensitivity is a major cause of model hallucination. Additionally, the authors conduct a series of experiments to validate the effectiveness of their proposed method.

**Strengths:**

1. The overall logic of the paper is sound. The authors theoretically analyze the phenomena of across-image insensitivity and within-image insensitivity in multimodal DPO, revealing that a significant part of the factors affecting DPO performance stems from visual insensitivity.

2. The experiments in the paper are sufficiently comprehensive and the presentation is relatively easy to follow.

**Weaknesses:**

The main weakness of this paper is the lack of substantive innovation. Although the authors provide a theoretical explanation of across-image insensitivity and within-image insensitivity, I fail to see a strong connection between these phenomena and the proposed method. The approach appears to be merely a combination of standard DPO and DPO with reject images. The statement "In summary, both types of issues arise from visual insensitivity" alone is insufficient to demonstrate the relevance of the proposed method to the two types of insensitivity. If the authors can adequately address this issue, I would consider raising my score.

**Questions:**

1. Please refer to the weaknesses outlined above.

2. The construction method for reject images proposed by the authors lacks innovation and relies heavily on the performance of CLIP. Would the effectiveness of the proposed approach be improved if image editing datasets were used to construct chosen-reject pairs?

---

> ### Author Response · Authors · 2025-11-23
> **Response to Reviewer kyXb (1/3)**
>
> We greatly appreciate your thoughtful feedback and insightful suggestions. Below, we address your concerns in detail, aiming to clarify and improve our paper.
>
> -----------------------------------
>
> Our work introduces the theoretical formalization of across-image and within-image insensitivity in multimodal DPO, proposes perception-enhanced preference data and a dual-margin objective (mPEA-DPO) directly derived from this analysis, and demonstrates consistent SOTA gains across hallucination benchmarks. We have added ${\color{blue}\text{Appendix C}}$ to explicitly connect these two failure modes with the image-level and response-level margins ($G\_{m}$, $G\_{r}$), and to show both theoretically and empirically how mPEA-DPO enlarges these margins. Additionally, we include new DINO-based construction experiments to verify encoder generalizability, and we further discuss the feasibility and limitations of constructing chosen–rejected pairs using image-editing datasets.
>
> **Weakness**
>
> > The main weakness of this paper is the lack of substantive innovation.
>
> We apologize that our original writing did not clearly convey the novelty of our contributions. To the best of our knowledge, our work is the first to:
>
> * **Theoretically formalize two fundamental failure modes of multimodal DPO — across-image and within-image insensitivity.** Existing multimodal DPO variants (e.g., mDPO, V-DPO, CHiP, DAMA) adapt the DPO objective but do not explicitly decompose the preference signal into visual and textual components or characterize these two distinct failure modes. To the best of our knowledge, we are the first to theoretically identify and formalize two fundamental failure modes in MLLMs.
>
> * **Design perception-enhanced preference data that is directly motivated by this analysis.** Our CLIP-guided construction specifically creates image pairs that maximize the visual preference gap, which our ablations show to be substantially more effective than standard perturbation strategies such as rotation, cropping, or black images.
>
> * **Introduce mPEA-DPO, a dual-preference objective derived from the above theory, which jointly optimizes response-level and image-level margins (($G\_r$, $G\_m$)) in a reference-free and margin-controlled way.** Ablation studies (Tables 2 and 3) demonstrate that both components are necessary and that our data- and objective-level designs together yield SOTA performance on three hallucination benchmarks.

---

> ### Author Response · Authors · 2025-11-23
> **Response to Reviewer kyXb (2/3)**
>
> **Weakness**
>
> > The statement "In summary, both types of issues arise from visual insensitivity" alone is insufficient to demonstrate the relevance of the proposed method to the two types of insensitivity.
>
> We agree that the original paper did not clearly articulate how the proposed mPEA-DPO objective is connected to the two forms of visual insensitivity. We have now added ${\color{blue}\text{Appendix C}}$ to make this link explicit.
>
>
> In short, starting from Definitions 1–2, we show that across-image insensitivity can be characterized by a small **image-level margin** $G\_m = \log \pi\_\theta(y\_w|x,m\_w)- \log \pi\_\theta(y\_w|x,m\_l),$ i.e., the model’s confidence in the preferred response does not increase when we replace a context-reduced image ($m\_l$) with the correct image ($m\_w$). Similarly, within-image insensitivity corresponds to a small **response-level margin** $G\_r = \log \pi\_\theta(y\_w|x,m\_w)- \log \pi\_\theta(y\_l|x,m\_w),$ meaning the model fails to distinguish the preferred answer from the dispreferred one even when conditioned on the correct image.
>
> The mPEA-DPO objective is then designed to **directly enlarge these two margins**: the mVPO term increases ($G\_m$) (image-level discrimination across ($m\_w$) and ($m\_l$)), while the mRPO term increases ($G\_r$) (response-level discrimination within ($m\_w$)). Therefore, optimizing mPEA-DPO theoretically mitigates both across- and within-image insensitivity. We provide full derivations and an empirical study of how ($G\_m$) and ($G\_r$) change before and after training in the new ${\color{blue}\text{Appendix C}}$.

---

> ### Author Response · Authors · 2025-11-23
> **Response to Reviewer kyXb (3/3)**
>
> **Question**
>
> >The construction method for rejected images relies heavily on the performance of CLIP.
>
> To evaluate the generalizability of our rejected image construction method, we also considered using DINOv2-base as the visual encoder for guiding the construction of rejected images. We conducted experiments on MMHal-Benchmark, Object HalBench, and AMBER, and the results are shown in the table below. We observe that both CLIP-based and DINO-based construction methods effectively reduce hallucination. This result indicates that our approach generalizes well across different types of visual encoders.
>
> We will clarify in the camera-ready that CLIP is chosen purely for convenience and that other encoders (e.g., DINOv2) can be plugged in without changing the framework.
>
> * MMHal-Bench:
>
> |  | Score | Hal. |
> |-------|-------|-------|
> | mPEA-DPO (DINO)| 3.02 | 0.36 |
> | mPEA-DPO (CLIP) | 2.96 | 0.35 |
>
> * Object HalBench:
>
> |  | CHAIRs | CHAIRi |
> |-------|-------|-------|
> | mPEA-DPO (DINO)| 4.7 | 3.0 |
> | mPEA-DPO (CLIP) | 4.3 | 3.2 |
>
> * AMBER:
>
> |  | C. | Cover | Hal. | Cog |
> |-------|-------|-------|-------|-------|
> | mPEA-DPO (DINO) | 2.2 | 46.6 | 10.5 | 0.7 |
> | mPEA-DPO (CLIP) | 1.9 | 46.7 | 10.3 | 0.6 |
>
> >Would the effectiveness of the proposed approach be improved if image editing datasets were used to construct chosen-rejected pairs?
>
> Using image editing datasets to construct high-quality chosen-rejected pairs is indeed an interesting idea. However, there are several limitations:
> * **Sample mismatch:** Image editing datasets typically contain samples of the form $(x,m,m\_e)$, where $x$ donates an editing instruction, $m$ donates the original image, $m\_e$ donates the edited image. In contrast, DPO typically requires $(q,m,y)$ samples, where $q$ is a prompt containing a question related to image $m$, and $y$ is the corresponding response. Therefore, existing image editing datasets lack the question-answer pairs required for direct DPO training.
> * **Computational overhead:** Generating chosen-rejected pairs via image editing would require precise editing models, which are generally much more computationally expensive than encoders like CLIP or DINO.
>
> Nonetheless, we agree that image editing is a promising avenue for creating high-quality chosen-rejected pairs, and we plan to explore this approach in future work.

---

> ### Author Response · Authors · 2025-11-27
> **Looking Forward to Your Feedback**
>
> Dear Reviewer kyXb,
>
> Thank you for your insightful review. We have added further experiments and analyses addressing all your points. If you have any additional suggestions or questions, please feel free to let us know.
>
> We appreciate your time and consideration.
>
> Best regards, The Authors

---

### Official Review · Reviewer_PATn · 2025-10-31

**Soundness:** 3
**Presentation:** 2
**Contribution:** 3
**Rating:** 6
**Confidence:** 4

**Summary:**

This work introduces Perception-Enhanced Alignment DPO (PEA-DPO). The proposed method uses both image-level and response-level feedback. The authors show that PEA-DPO reduces hallucinations consistently for multiple base models and model scales.

**Strengths:**

* The concepts of across-image and within-image insensitivity are well motivated and the mathematical background is both intuitive and presented in an accessible manner.
* The method is described clearly and appears easily reproducible.
* The reported results on hallucination benchmarks are very strong.

**Weaknesses:**

* In table 1, the main results, some numbers and details appear inconsistent with previously published results. For example, POVID reports an MMHalBench number of 2.69 [3], but the table shows 2.08. The table lists LLaVA-RLHF as building on LLaVA-v1.5-7B but the paper describes building on LLaVA with a different SFT recipe [4]. These are some initial discrepancies that I noticed, but due to time constraints I am unable to review every reported result. Instead, I would urge the authors to carefully review the reported results and details shown in table 1 for a fair comparison of this work’s contributions as part of the rebuttal period.
* The manuscript primarily reports hallucination metrics, with only one metric, MMMU, to speak to the model’s task performance post-alignment. On MMMU, the method reports some regressions across both model scales under test. Evaluation on quality-focused benchmarks such as MMStar, MME, AI2D, or MMVet would give a better understanding on the impact on helpfulness.
* As described in [5], CHAIR scores suffer from not penalizing shorter responses to avoid hallucinations. The paper reports a reduction in coverage as reported in AMBER, which is also briefly discussed in 5.2. While the reduction in CHAIR metrics is very strong for this method, without reporting recall, it is hard to interpret whether this is strictly due to a reduction of hallucinations or perhaps more trivially by producing shorter / potentially less informative responses, also related to the previous point.
* In table 1, the commercial baseline reported is GPT-4V. A more recent frontier model may better put the reported results in perspective.
* The manuscript could more strongly discuss the natural connection to mDPO [2]. For example, the proposed VPO objective is identical to the “conditional preference optimization” proposed in mDPO. and the $L_{PEA-DPO}$ is identical to $L_{mDPO}$ without their $L_{AncPO}$ anchoring term. Similarly, the ablation on cropping in 5.5 makes the method more similar to mDPO, yet results remain much stronger than mDPO, which may also benefit from some discussion.


[1] Tong, Shengbang, et al. "Eyes wide shut? exploring the visual shortcomings of multimodal llms." Proceedings of the IEEE/CVF Conference on Computer Vision and Pattern Recognition. 2024.

[2] Wang, Fei, et al. "mdpo: Conditional preference optimization for multimodal large language models." arXiv preprint arXiv:2406.11839 (2024).

[3] Zhou, Yiyang, et al. "Aligning modalities in vision large language models via preference fine-tuning." arXiv preprint arXiv:2402.11411 (2024).

[4] Sun, Zhiqing, et al. "Aligning large multimodal models with factually augmented rlhf." arXiv preprint arXiv:2309.14525 (2023).

[5] Amirloo, Elmira, et al. "Understanding alignment in multimodal llms: A comprehensive study." arXiv preprint arXiv:2407.02477 (2024).

[6] Chen, Lin, et al. "Are we on the right way for evaluating large vision-language models?." Advances in Neural Information Processing Systems 37 (2024): 27056-27087.

**Questions:**

* The construction of perception-enhanced preference data is using CLIP to identify when crucial detail has been removed, though many popular multimodal models are trained with encoders following CLIP-style pre-training (or similar) so perhaps corruptions that lead to difference in CLIP representations may be most likely to lead to perceptible differences for the multimodal LLM under alignment. [1] shows that different encoders can have different “blind spots”. Have you considered using a different encoder (DINO?), or perhaps an ensemble to determine when critical visual context has been removed?

---

> ### Author Response · Authors · 2025-11-22
> **Response to Reviewer PATn (1/3)**
>
> We thank the reviewer for their thoughtful feedback and constructive suggestions. Below, we address your comments in detail.
>
> ----------------------
>
> **Weakness**
>
> **W1: Carefully review the reported results and details shown in Table 1 for a fair comparison of this work’s contributions.**
>
> R1: We apologize for not clearly specifying in the manuscript how the results in Table 1 were obtained. We have carefully followed the publicly available code and released checkpoints to reproduce these results, aiming to provide a fair comparison. Evaluating MMHalBench and Object HalBench requires querying the OpenAI API. To ensure a fair and consistent comparison, we follow Rlaif-v[1] and use **gpt-4-1106-preview** for MMHalBench and **gpt-3.5-turbo-0613** for Object HalBench.
>
> * or the hallucination-specific baselines, including VCD, OPERA, HALC, and EOS, we reproduced the results using the authors’ publicly released code on the LLaVA-v1.5 model.
>
> * For the RLHF/RLAIF-based baselines, we used the official code and checkpoints provided by the authors. Specifically, for LLaVA-RLHF, HA-DPO, HALVA, RLAIF-V, OPA-DPO, DAMA, and RLHF-V, we directly evaluated their publicly released checkpoints, while for mDPO, and POVID, we reproduced the results by running the authors’ publicly available code.
>
> **W2: Evaluation on quality-focused benchmarks would give a better understanding on the impact on helpfulness.**
>
> R2: In addition to MMMU, we evaluated the general capabilities of mPEA-DPO on two quality-focused benchmarks, MMStar and AI2D, and the results are presented in the table below. We observe that mPEA-DPO outperforms its backbone, LLaVA-v1.5-7B, on both MMStar and AI2D, and its performance on MMMU shows only negligible differences compared with the backbone.
> |  | MMStar | AI2D | MMMU |
> |-------|-------|-------|-------|
> | LLaVA-v.15-7B | 30.3 | 49.1 | 35.3 |
> | mPEA-DPO (Ours) | 32.8 | 51.9 | 33.4 |
>
> Furthermore, as shown in the following table, we provide a breakdown of results across the six core capabilities in MMStar, with each category containing 250 samples. These results further demonstrate that our method does not compromise the model’s general capabilities; rather, the model exhibits improved general performance after post-alignment.
>
> CP (coarse perception), FP (fine-grained perception), IR(instance reasoning), LR (logical reasoning), ST (science & technology), MA (mathematics).
> |  | CP | FP | IR | LR | ST | MA | AVG|
> |-------|-------|-------|-------|-------|-------|-------|-------|
> | LLaVA-v.15-7B | 58.8 | 24.0 | 38.8 | 24.0 | 13.6 | 22.8 | 30.3 |
> | mPEA-DPO (Ours) | 62.4 | 30.0 | 39.6 | 23.6 | 24.0 | 17.2 | 32.8 |
>
> **W3: While the reduction in CHAIR metrics is very strong for this method, without reporting recall, it is hard to interpret whether this is strictly due to a reduction of hallucinations.**
>
> R3: To demonstrate that the reduction in CHAIR metrics is indeed driven by reduced hallucination rather than shorter or less informative outputs, we conducted a comparison on Object HalBench between our method and two of the strongest existing baselines, OPA-DPO[2] and DAMA[3]. We report three metrics: CHAIRs, CHAIRi, and Recall.
>
> As shown in the table below, mPEA-DPO achieves clearly superior hallucination metrics compared with both OPA-DPO and DAMA. Importantly, mPEA-DPO attains recall performance comparable to DAMA while substantially outperforming OPA-DPO, indicating that the improvement in CHAIR metrics is not simply due to shorter responses but rather reflects a genuine reduction in hallucination.
> |  | CHAIRs | CHAIRi | Recall |
> |-------|-------|-------|-------|
> |OPA-DPO | 13.3 | 4.3 | 43.29 |
> | DAMA | 10.3 | 5.9 | 54.50 |
> | mPEA-DPO (Ours) | 4.3 | 3.2 | 52.43 |
>
> [1]Yu, Tianyu, et al., Rlaif-v: Open-source ai feedback leads to super gpt-4v trustworthiness. CVPR 2025.
>
> [2] Yang, Zhihe, et al., Mitigating Hallucinations in Large Vision-Language Models via DPO: On-Policy Data Hold the Key. CVPR 2025.
>
> [3] Lu, Jinda, et al., DAMA: Data- and Model-aware Alignment of Multi-modal LLMs. ICML 2025.

---

> ### Author Response · Authors · 2025-11-22
> **Response to Reviewer PATn (2/3)**
>
> **Weakness**
>
> **W4: The commercial baseline reported is GPT-4V. A more recent frontier model may better put the reported results in perspective.**
>
> R4: We have additionally considered a more recent commercial baseline, Gemini-2.5-Pro, which was released in June of this year. We evaluated it on three hallucination benchmarks, MMHal Bench, Object HalBench, and AMBER, and the results are presented in the table below.
> We observe that the response quality metrics, such as the overall Score, of mPEA-DPO are comparable to those of Gemini 2.5 Pro. At the same time, mPEA DPO achieves clearly better hallucination metrics on all three benchmarks. This provides further evidence that our method effectively reduces hallucinations while maintaining strong response quality.
>
> * MMHal-Bench:
>
> |  | Score | Hal. |
> |-------|-------|-------|
> | Gemini-2.5-pro | 3.55 | 0.28 |
> | mPEA-DPO, 13B (Ours) | 3.16 | 0.31 |
>
> * Object HalBench:
>
> |  | CHAIRs | CHAIRi |
> |-------|-------|-------|
> | Gemini-2.5-pro | 12.0 | 8.2 |
> | mPEA-DPO, 13B (Ours) | 4.3 | 2.7 |
>
> * AMBER:
>
> |  | C. | Cover | Hal. | Cog |
> |-------|-------|-------|-------|-------|
> | Gemini-2.5-pro | 9.5 | 78.0 | 75.1 | 5.2 |
> | mPEA-DPO, 13B (Ours) | 2.9 | 50.0 | 12.8 | 0.8 |

---

> ### Author Response · Authors · 2025-11-25
> **Response to Reviewer PATn (3/3)**
>
> **W5: The manuscript could more strongly discuss the natural connection to mDPO**
>
> R5: We apologize for not clearly explaining in the manuscript how mPEA-DPO differs from mDPO. We would like to clarify that mPEA-DPO and mDPO differ in four main aspects. Furthermore, to provide a fair comparison, we evaluated mDPO and mPEA-DPO under the same training and evaluation settings.
>
> 1. **The optimization objectives differ.**
>
> The optimization objective of mPEA-DPO is as follows:
>
> $$
> \mathcal{L}\_{\text{mPEA-DPO}}=\mathrm{ReLU}\Bigg[ -\bigg( \frac{\log \pi\_{\theta}(y\_{w}|x, m\_{w})}{|y\_{w}|} - \frac{\log \pi\_{\theta}(y\_{l}|x, m\_{w})}{|y\_{l}|} - \gamma\_{r} \Bigg)\Bigg] \\\\ + \alpha\cdot\mathrm{ReLU}\Bigg[ -\bigg( \frac{\log \pi\_{\theta}(y\_{w}|x, m\_{w})}{|y\_{w}|} - \frac{\log \pi\_{\theta}(y\_{w}|x, m\_{l})}{|y\_{w}|} - \gamma\_{m}\Bigg)\Bigg]
> $$
>
> The optimization objective of mDPO is given by:
>
> $$
> \mathcal{L}\_{\text{mDPO}}=-\log\sigma\Bigg(\beta\frac{\log \pi\_{\theta}(y\_{w}|x, m\_{w})}{\log \pi\_{\text{ref}}(y\_{w}|x, m\_{w})} - \beta\frac{\log \pi\_{\theta}(y\_{l}|x, m\_{w})}{\log \pi\_{\text{ref}}(y\_{l}|x, m\_{w})}\Bigg) \\\\ -\log\sigma\Bigg(\beta\frac{\log \pi\_{\theta}(y\_{w}|x, m\_{w})}{\log \pi\_{\text{ref}}(y\_{w}|x, m\_{w})} - \beta\frac{\log \pi\_{\theta}(y\_{w}|x, m\_{l})}{\log \pi\_{\text{ref}}(y\_{w}|x, m\_{l})}\Bigg) \\\\ -\log\sigma\Bigg(\beta\frac{\log \pi\_{\theta}(y\_{w}|x, m\_{w})}{\log \pi\_{\text{ref}}(y\_{w}|x, m\_{w})} - \delta\Bigg)
> $$
>
> From these objectives, it is evident that mDPO follows a form similar to the DPO formulation. In Section 4.2 of our manuscript, we detail three key modifications introduced in mPEA-DPO compared to standard DPO. These modifications include: (1) the removal of the reference model, (2) the replacement of the Sigmoid function with the ReLU function, and (3) the introduction of the hyperparameters $\gamma\_{r}$ and $\gamma\_{m}$ while removing $\beta$.
>
> 2. **The construction of rejected images differs.**
>
> mDPO uses random cropping to generate rejected images, whereas our method uses CLIP guidance to produce perception-enhanced rejected images.
>
> 3. **The training datasets differ.**
>
> mDPO is trained on the Silkie dataset, which contains 10k preference pairs. Following the practice of DAMA and OPA-DPO, we use the RLAIF-V dataset, which contains 22k preference pairs.
>
> 4. **The training configurations differ.**
>
> Following DAMA, we adopt full parameter finetuning. In contrast, mDPO uses LoRA finetuning.
>
> To ensure a fair comparison between mPEA-DPO and mDPO, we apply both methods to LLaVA-v1.5-7B using the same training data and the same training configuration, and we evaluate them on MMHal Bench and AMBER. In addition, to directly address the reviewer’s concern regarding the cropping ablation in Section 5.5, we replace the CLIP-guided rejected image construction with random cropping and denote this ablation variant as mPEA-DPO (Crop). The results are shown in the table provided below.
>
> * MMHal-Bench:
>
> |  | Score | Hal. |
> |-------|-------|-------|
> | mDPO | 2.39 | 0.52 |
> | mPEA-DPO (Crop) | 2.69 | 0.42 |
> | mPEA-DPO | 3.02 | 0.36 |
>
> * AMBER:
>
> |  | C. | Cover | Hal. | Cog |
> |-------|-------|-------|-------|-------|
> | mDPO | 4.7 | 49.5 | 21.4 | 2.2 |
> | mPEA-DPO (Crop) | 2.1 | 46.8 | 11.3 | 0.9 |
> | mPEA-DPO | 1.9 | 46.7 | 10.3 | 0.6 |
>
> We draw the following conclusions from the results.
> (1) Under identical training data and training configurations, mPEA-DPO improves both response quality and hallucination metrics compared with mDPO.
> (2) Although mPEA-DPO (Crop) uses the same preference data as mDPO, it still outperforms mDPO on both MMHal-Bench and AMBER. This improvement arises from our novel optimization objective and demonstrates the advantage of mPEA-DPO in mitigating hallucination.
>
> **Question**
>
> **Q1: Have you considered using a different encoder (DINO?), or perhaps an ensemble to determine when critical visual context has been removed?**
>
> A1: We have investigated the use of DINOv2-base as an alternative visual encoder to guide the construction of rejected images. We conducted experiments on MMHal-Bench, Object HalBench, and AMBER, and the results are presented in the table below. We find that using either CLIP or DINOv2 to guide the construction of rejected images effectively reduces hallucination. This observation indicates that our approach generalizes well across different visual encoders and is not dependent on a specific choice of encoder.
>
> * MMHal-Bench:
>
> |  | Score | Hal. |
> |-------|-------|-------|
> | mPEA-DPO (DINO)| 3.02 | 0.36 |
> | mPEA-DPO (CLIP) | 2.96 | 0.35 |
>
> * Object HalBench:
>
> |  | CHAIRs | CHAIRi |
> |-------|-------|-------|
> | mPEA-DPO (DINO)| 4.7 | 3.0 |
> | mPEA-DPO (CLIP) | 4.3 | 3.2 |
>
> * AMBER:
>
> |  | C. | Cover | Hal. | Cog |
> |-------|-------|-------|-------|-------|
> | mPEA-DPO (DINO) | 2.2 | 46.6 | 10.5 | 0.7 |
> | mPEA-DPO (CLIP) | 1.9 | 46.7 | 10.3 | 0.6 |

---

> ### Author Response · Authors · 2025-11-27
> **Looking Forward to Your Feedback**
>
> Dear Reviewer PATn,
>
> Thank you for your insightful review. We have added further experiments and analyses addressing all your points. If you have any additional suggestions or questions, please feel free to let us know.
>
> We appreciate your time and consideration.
>
> Best regards, The Authors

---

### Author Response · Authors · 2025-12-02
**Summaried Comment by Authors**

We sincerely thank the Area Chairs for evaluating our submission. We also thank all reviewers for their valuable and insightful feedback.

The following text will (1) summarize the core contributions and significance of PEA-DPO, along with the strengths highlighted by the reviewers, and (2) provide a concise overview of the key clarifications and additional analyses introduced during the rebuttal.

## **Highlights of Contributions:**

* **Theoretically formalize two fundamental failure modes of multimodal DPO — across-image and within-image insensitivity.** Existing multimodal DPO variants (e.g., mDPO, V-DPO, CHiP, DAMA) adapt the DPO objective but do not explicitly decompose the preference signal into visual and textual components or characterize these two distinct failure modes. To the best of our knowledge, we are the first to identify and formalize two fundamental failure modes in MLLMs theoretically. Especially, Reviewers $({\color{red}\text{PATn}}$, ${\color{orange}\text{KtuX}})$ appreciated that our formulation of **across-image** and **within-image insensitivity** is well motivated, and that the mathematical background is intuitive and presented in an accessible manner, and Reviewer ${\color{green}\text{KyXb}}$ further agreed that our theoretical analysis reveals that a significant part of the factors affecting DPO performance stems from visual insensitivity.

* **Design perception-enhanced preference data that is directly motivated by this analysis.** Our CLIP-guided construction specifically creates image pairs that maximize the visual preference gap, which our ablations show to be substantially more effective than standard perturbation strategies such as rotation, cropping, or black images. Especially, Reviewers $({\color{red}\text{PATn}},{\color{green}\text{KyXb}})$ appreciated that our proposed method is described clearly, appears easily reproducible, and is easy to follow.

* **Introduce mPEA-DPO, a dual-preference objective derived from the above theory, which jointly optimizes response-level and image-level margins (($G\_r$, $G\_m$)) in a reference-free and margin-controlled way.** Ablation studies (Tables 2 and 3) demonstrate that both components are necessary and that our data- and objective-level designs together yield SOTA performance on three hallucination benchmarks. Especially, Reviewers $({\color{red}\text{PATn}}, {\color{orange}\text{KtuX}})$ acknowledged the strong performance of mPEA-DPO on hallucination benchmarks, and Reviewers $({\color{green}\text{KyXb}}, {\color{orange}\text{KtuX}})$ agreed that the experiments presented in our manuscript are sufficiently comprehensive.

## **Summary of Rebuttal:**

We truly appreciate the constructive suggestions and critiques, which helped us identify several important areas for improvement. Below we summarize the main points of discussion, followed by clarifications and detailed responses to each reviewer:

* **Reviewer ${\color{red}\text{PATn}}$**: We incorporated additional evaluation benchmarks, including MMStar and AI2D, and added up-to-date commercial models such as Gemini-2.5-Pro as baselines. We also reported complementary metrics (e.g., recall on Object HalBench) to validate the effectiveness of mPEA-DPO further.

* **Reviewer ${\color{green}\text{KyXb}}$**: We extended the theoretical analysis to more explicitly connect mPEA-DPO with the two identified issues, namely **within-image insensitivity** and **across-image insensitivity**, and added detailed experiments showing that mPEA-DPO can effectively mitigate both. The reviewer indicated that clarifying this connection was important for their overall assessment, and we provide additional theoretical discussion and supporting empirical analyses in ${\color{blue}\text{Appendix C}}$ to help address this concern. In addition, we included experiments demonstrating that alternative visual encoders (e.g., DINO) can be seamlessly integrated into our framework without modification.

* **Reviewer ${\color{orange}\text{KtuX}}$:** We conducted additional empirical studies, including varying the number of candidate masks, the masking ratio, and the similarity metrics, to validate their relationship with mPEA-DPO. Moreover, for a fully fair comparison, we evaluated mPEA-DPO alongside competitive baselines under exactly the same training and evaluation settings, and reported both performance and computational cost.

As the author–reviewer discussion period concludes, we sincerely hope that our revisions and clarifications adequately address your concerns. Should any questions remain, we would be more than happy to provide further explanations. Once again, we are truly grateful to all reviewers for their thoughtful comments and for helping us improve the quality of our work.

---

> ### Author Response · Authors · 2025-12-02
> **Updated Manuscript**
>
> We sincerely thank all reviewers for their thorough and valuable feedback. In response to the comments, we have added several experiments and analyses. While ensuring full consistency with the original contributions and conclusions, we made targeted revisions to enhance completeness and clarity. For ease of reference, all modifications are highlighted in ${\color{blue}\text{blue}}$ in the manuscript. A brief summary of the major updates is provided below:
>
> | Changes | Section | Related Reviewers |
> |-------|-------|-------|
> | sources for reported results and details in Table 1 | Section 5 | ${\color{red}\text{PATn}}$ |
> | a new baseline using the latest commercial model | Section 5 | ${\color{red}\text{PATn}}$ |
> | theoretical and empirical analyses connecting mPEA-DPO to the two insensitivity issues | Appendix C | ${\color{green}\text{KyXb}}$ |
> | hyperparameter analysis | Appendix D | ${\color{red}\text{PATn}}$ |
> | experiments on generalization across different visual encoders | Appendix E | ${\color{red}\text{PATn}}$, ${\color{green}\text{KyXb}}$, ${\color{orange}\text{KtuX}}$ |
> | recall reporting on Object HalBench | Appendix F | ${\color{red}\text{PATn}}$, ${\color{orange}\text{KtuX}}$ |
> | analysis of the effect of mVPO on general capabilities | Appendix G | ${\color{orange}\text{KtuX}}$ |
> | fair comparisons against competitive baselines | Appendix H | ${\color{orange}\text{KtuX}}$ |
> | additional general capability experiments and analyses | Appendix I | ${\color{red}\text{PATn}}$, ${\color{orange}\text{KtuX}}$ |

---

### Meta-Review · Area_Chair_Gi7Z · 2025-12-26

**Summary:**

This paper identifies a failure mode in multimodal DPO: visual insensitivity, with across-image and within-image manifestations. The authors propose PEA-DPO, a dual-preference objective that optimizes both response-quality preferences and visual-context preferences. It received scores of 466. Reviewers agree that the method is described clearly, the concepts of across-image and within-image insensitivity are well motivated, and the reported results on hallucination benchmarks are strong. However, several major concerns still remain, regarding limited evaluation benchmarks and the use of old backbones to conduct experiments. Overall, the rebuttal is not convincing enough, and the AC would like to recommend rejection.

**Reviewer Concerns:**

Concerns adequately addressed:

1. The authors added up-to-date commercial models such as Gemini-2.5-Pro as baselines, and also reported complementary metrics (e.g., recall on Object HalBench) to validate the effectiveness of mPEA-DPO further.

2. The authors included experiments demonstrating that alternative visual encoders (e.g., DINO) can be integrated into their framework as well.

3. The authors extended the theoretical analysis to more explicitly connect mPEA-DPO with the two identified issues.

4. The authors conducted additional empirical studies, including varying the number of candidate masks, the masking ratio, and the similarity metrics, to validate their relationship with mPEA-DPO.


Concerns insufficiently addressed:

1. The reliance on older backbones (LLaVA-1.5 7B and 13B) remains a limitation. To ensure stronger and more reliable conclusions, the authors should evaluate their method on modern backbones such as Qwen2.5-VL. Given current standards, the AC thinks that LLaVA-1.5 is no longer competitive and cannot fully support claims of general effectiveness.

2. The evaluation scope is still narrow. The benchmarks are primarily hallucination-focused and grounded in natural images. Although MMStar and AI2D results were added, the overall coverage remains insufficient, and broader evaluation would be necessary for a more comprehensive assessment to understand the impact of reducing hallucination to other multimodal benchmarks.

**Reviewer Scores:**

The AC thinks that the rebuttal is not convincing enough for the reviewers to increase the scores.

---

### Decision · Program_Chairs · 2026-01-26

Reject